



# Intercomparison of wind observations from ESA's satellite mission Aeolus and the ALADIN Airborne Demonstrator

Oliver Lux[1], Christian Lemmerz[1], Fabian Weiler[1], Uwe Marksteiner[1], Benjamin Witschas[1], Stephan Rahm[1], Alexander Geiss[2], Oliver Reitebuch[1]

[1] German Aerospace Center (Deutsches Zentrum für Luft- und Raumfahrt e.V., DLR), Institute of Atmospheric Physics, Oberpfaffenhofen 82234, Germany
[2] Ludwig-Maximilians-University Munich, Meteorological Institute, 80333 Munich, Germany

*Correspondence to*: Oliver Lux (oliver.lux@dlr.de)

**Abstract.** Shortly after the successful launch of ESA's wind mission Aeolus, carried out by the European Space Agency, collocated airborne wind lidar observations were performed in Central Europe, employing the prototype of the satellite instrument, the ALADIN Airborne Demonstrator (A2D). Like the direct-detection Doppler wind lidar on-board Aeolus, the A2D is composed of a frequency-stabilised ultra-violet laser, a Cassegrain telescope and a dual-channel receiver to measure line-of-sight (LOS) wind speeds by analysing both Mie and Rayleigh backscatter signals. In the frame of the first airborne validation campaign after the launch still during the commissioning phase of the mission, four coordinated flights along the satellite swath were conducted in late autumn of 2018, yielding wind data in the troposphere with high coverage of the Rayleigh channel. Owing to the different measurement grids and viewing directions of the satellite and airborne instrument, intercomparison with the Aeolus wind product requires adequate averaging as well as conversion of the measured A2D LOS wind speeds to the satellite LOS. The statistical comparison of the two instruments with model wind data from the ECMWF shows biases of the A2D and Aeolus LOS wind speeds of -0.9 m·s$^{-1}$ and +1.6 m·s$^{-1}$, respectively, while the random errors are around 2.5 m·s$^{-1}$. The paper also discusses the influence of different threshold parameters implemented in the comparison algorithm as well as an optimization of the A2D vertical sampling to be used in forthcoming validation campaigns.

## 1 Introduction

On 22 August 2018, the fifth Earth Explorer mission of the European Space Agency (ESA) – Aeolus – has been launched to space, marking an important milestone in the centennial history of atmospheric observing systems (Stith et al., 2018; Kanitz et al., 2019; Reitebuch et al., 2019; Straume et al., 2019). Aeolus is the first mission to acquire atmospheric wind profiles from the ground to the lower stratosphere on a global scale deploying the first-ever satellite-borne wind lidar system ALADIN (Atmospheric LAser Doppler INstrument) (ESA, 2008; Stoffelen et al., 2005; Reitebuch, 2012). Circling the Earth on a sun-synchronous orbit with a repeat cycle of one week, ALADIN provides one component of the wind vector along the instrument's line-of-sight (LOS) from ground up to 30 km altitude with a vertical resolution of 0.25 km to 2 km depending on altitude. The near-real-time wind observations from Aeolus contribute to improving the accuracy of numerical weather


prediction (Rennie and Isaksen, 2019a; Isaksen and Rennie, 2019; Rennie and Isaksen, 2019b) and advance the understanding of atmospheric dynamics and processes relevant to climate variability. In particular, wind profiles acquired in the tropics and over the oceans help to close large gaps in the global wind data coverage which, before the launch of Aeolus, represented a major deficiency in the Global Observing System (Baker et al., 2014; Andersson, 2018; NAS, 2018). In

addition to the wind data product, Aeolus provides information on cloud top heights and on the vertical distribution of aerosol and cloud properties such as backscatter and extinction coefficients (Flamant et al., 2008; Ansmann et al., 2007).

Already several years before the satellite launch, an airborne prototype of the Aeolus payload, the ALADIN Airborne Demonstrator (A2D), was developed at the German Aerospace Center (Deutsches Zentrum für Luft- und Raumfahrt e.V., DLR). Due to its representative design and operating principle, the A2D has since delivered valuable information on the

wind measurement strategies of the satellite instrument as well as on the optimization of the wind retrieval and related quality-control algorithms. Broad vertical and horizontal coverage across the troposphere is achieved thanks to the complementary design of the A2D receiver which, like ALADIN, comprises a Rayleigh and a Mie channel for analysing both molecular and particulate backscatter signals. In addition to the A2D, a well-established coherent Doppler wind lidar (2-μm DWL) has been operated at DLR for many years. Being equipped with a double-wedge scanner the 2-μm DWL

allows determining the wind vector with high accuracy and precision (Weissmann et al., 2005; Chouza et al., 2016; Witschas et al., 2017). Both wind lidar systems thus represent key instruments for the calibration and validation (Cal/Val) activities during the Aeolus mission.

Over the past years, both systems were deployed in several field experiments for the purpose of pre-launch validation of the satellite instrument and of performing wind lidar observations under various atmospheric conditions (Marksteiner et al.,

2018; Lux et al., 2018). In autumn of 2018, the first airborne campaign after the launch of Aeolus was carried out from the airbase in Oberpfaffenhofen, Germany. Aside from extending the existing data set of wind observations, this field experiment aimed to perform several underflights of Aeolus in Central Europe in order to provide first comparative wind result between the airborne demonstrator and the satellite instrument during its commissioning phase. Moreover, the campaign had the objective to optimize the operational procedures, particularly in terms of flight planning, to be applied

during the forthcoming Cal/Val campaigns in the operational phase of Aeolus.

This paper presents the results from the first airborne validation campaign of the Aeolus mission and demonstrates the methodology used to compare the different data sets from the airborne and satellite instrument. In this context, it intends to serve as a reference for later studies related to the airborne validation of Aeolus. The text is organized as follows. First, the design and operation principle of ALADIN and the airborne demonstrator are briefly described with a focus on the

commonalities and differences of the two wind lidar instruments (chapter 2). The subsequent section gives an overview of the validation campaign including the flight planning procedures and A2D calibration. Afterwards, the wind observations from the research flight along the satellite swath performed on 22 November 2018 are presented (sections 3.2, 3.3), followed by an assessment of the A2D wind data accuracy and precision by means of the 2-μm coherent wind lidar (section 3.4). In a next step, the adaptation of the A2D wind data to the Aeolus measurement grid and viewing geometry is explained (sections





4.1, 4.2), which is prerequisite for the subsequent comparison of the two data sets with each other and with model wind data from the ECMWF (section 4.3). The influence of two selected threshold parameters incorporated in the comparison algorithm on the outcome of the statistical comparison is discussed as well (section 4.4). Due to the sparse coverage of Mie wind data gained during the campaign, the analysis is restricted to the A2D and Aeolus Rayleigh channels. The comparison of the 2-µm DWL wind data with those of Aeolus is the subject of another publication (Witschas et al., 2019). Finally, an

optimized range gate setting of the A2D is proposed which aims to improve the validation capabilities of the instrument in forthcoming airborne campaigns to be conducted during the Aeolus mission (section 4.5).

## 2 ALADIN and its airborne demonstrator

The single payload of the Aeolus satellite, ALADIN, represents one of the most sophisticated instruments ever put into orbit. While it has been operating in space since its launch in August 2018, the ALADIN Airborne Demonstrator has already been

employed on ground and in research flights since 2005. The design and measurement principle of the two direct-detection Doppler wind lidars have been extensively specified in previous publications, describing the satellite (ESA, 2008; Stoffelen et al., 2005; Reitebuch, 2012; Kanitz et al., 2019) and airborne instrument (Reitebuch et al., 2009; Paffrath et al., 2009; Lux et al., 2018), respectively. Therefore, only a reduced description of the A2D is presented in this section, followed by a short explanation of the Aeolus wind data product that was validated later in the text.

**2.1 The A2D direct-detection wind lidar system**

A simplified schematic of the airborne instrument is illustrated in Fig. 1. Like ALADIN, the system consists of a pulsed, frequency-stabilised, ultra-violet (UV) laser transmitter, a Cassegrain-type telescope, a configuration to combine a fraction of the emitted radiation with the atmospheric and ground return signals (front optics), and a dual-channel receiver including detectors.

The laser transmitter is realized by a frequency-tripled Nd:YAG master oscillator power amplifier (MOPA) system which generates UV laser pulses at 354.89 nm wavelength with duration of 20 ns (full width at half maximum (FWHM)) and energy of 60 mJ at 50 Hz repetition rate (3.0 W average power). Injection-seeding of the master oscillator in combination with an active frequency stabilization technique (Lemmerz et al., 2017) provides single-frequency operation with a pulse-to-pulse frequency stability of approximately 3 MHz (rms) and a spectral bandwidth of 50 MHz (FWHM). The near-

diffraction-limited beam ($M^2 < 1.3$) is transmitted to the atmosphere via a piezo-electrically controlled mirror that is attached to the frame of a telescope in Cassegrain configuration. In contrast to the satellite instrument which uses a 1.5 m diameter telescope in transceiver configuration and operates at an off-nadir pointing angle of 35°, the A2D incorporates a 0.2 m diameter telescope which is oriented at an off-nadir angle of 20°. Owing to the structural design of the telescope, a range-dependent overlap function has to be considered in the wind retrieval as described in (Paffrath et al., 2009).


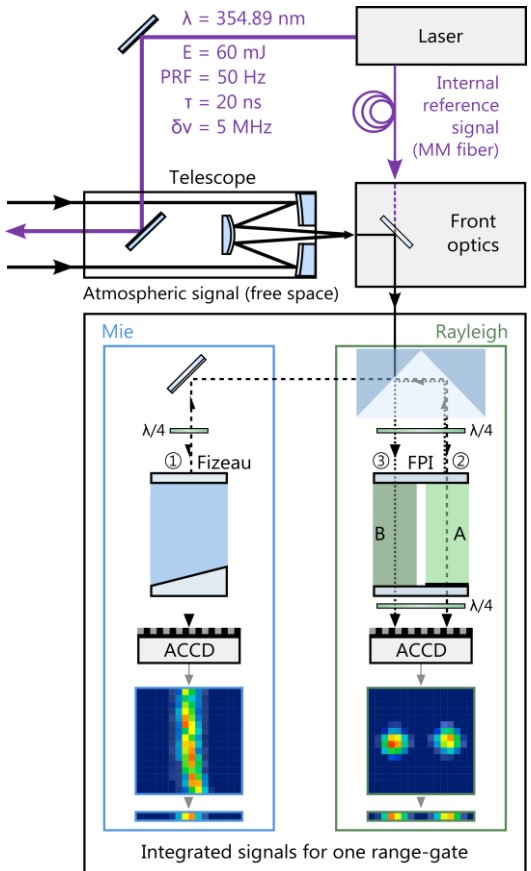

**Figure 1.** Schematic of the ALADIN Airborne Demonstrator (A2D) wind lidar instrument consisting of a frequency-stabilised, ultra-violet laser transmitter, a Cassegrain telescope, front optics and a dual-channel receiver. The latter is composed of a Fizeau interferometer and sequential Fabry-Pérot interferometers (FPI) for analysing the Doppler frequency shift from particulate and molecular backscatter signals, respectively. PRF: pulse repetition frequency, MM: multimode, ACCD: accumulation charge coupled device.

The backscattered radiation from the atmosphere and the ground is collected by the convex spherical secondary mirror of the telescope and directed to the front optics of the A2D receiver assembly. After passing through a narrowband UV bandpass filter (FWHM: 1.0 nm) which blocks the broadband solar background spectrum, the return signal is spatially overlapped with a small portion of the outgoing laser radiation which is referred to as internal reference signal. The latter is analysed for determining the transmitted laser frequency before the atmospheric return and for calibrating the frequency-dependent transmission of the receiver spectrometers which are required for accurate wind retrieval. Unlike Aeolus where the internal reference signal is guided to the front optics on free optical path, a multimode fiber (200 μm core diameter) is employed in the A2D. Utilization of the multimode fiber introduces detrimental speckle noise affecting the precision of the internal reference frequency determination, as explained in Lux et al. (2018). Hence, a fiber scrambler was recently integrated between the laser transmitter and the front optics in order to reduce the speckle noise and, in turn, to significantly improve the stability of the internal reference frequency and signal intensity (Lux et al., 2019).



The design of the A2D receiver is almost identical to that of the satellite instrument comprising two complementary channels to separately analyse the return signals from both molecules (Rayleigh channel) and particles like clouds and aerosols (Mie channel) (see lower part of Fig. 1). A Fizeau interferometer is used for measuring the Doppler frequency shift of the narrowband Mie signal (FWHM ≈ 50 MHz) that originates from cloud and aerosol backscattering, while two sequential

Fabry-Pérot interferometers (FPIs) are employed for determining the Doppler shift of the broadband (FWHM ≈ 3.8 GHz at 355 nm and 293 K) Rayleigh backscatter signal from molecules. The Mie channel is based on the fringe-imaging technique (McKay, 2002) which relies on the measurement of the spatial location of a linear interference pattern (fringe) that is vertically imaged onto the detector. A Doppler frequency shift $\Delta f_{\mathrm{Doppler}} = 2 f_0/c \cdot v_{\mathrm{LOS}}$ of the return signal ($f_0 = 844.75$ THz is the laser emission frequency, $c$ is the speed of light) manifests as a spatial displacement of the fringe centroid position with

an approximately linear relationship for typical wind speeds $v_{\mathrm{LOS}}$ along the laser beam LOS well below 100 m·s$^{-1}$ ($\Delta f_{\mathrm{Doppler}} <$ 563 MHz).

Due to the much broader spectral bandwidth of the molecular backscatter signal, a different technique is applied for deriving the Doppler frequency shift in the Rayleigh channel. Here, the measurement principle is based on the double-edge technique (Chanin et al., 1989; Garnier and Chanin, 1992; Flesia and Korb, 1999; Gentry et al., 2000) involving two bandpass filters

(A and B) which are realized by the sequential FPIs. The interferometers are specified such that the two transmission curves (free spectral range (FSR): 10.95 GHz, FWHM: 1.78 GHz, spacing: 6.18 GHz) are placed symmetrically around the frequency of the emitted laser pulse while the maxima are close to the inflexion points (edges) of the backscatter spectrum which exhibits a spectrally broadened Rayleigh-Brillouin line shape (Witschas et al., 2010; Witschas, 2011a; Witschas, 2011b). The signals $I_A$ and $I_B$ transmitted through the two bandpass filters are proportional to the convolution of the line

shape function of the atmospheric backscatter signal with the respective filter transmission function. Hence, measurement of the contrast between $I_A$ and $I_B$ ($(I_A - I_B)/(I_A + I_B)$) allows determining the frequency shift between the emitted and backscattered laser pulse.

Detection of the Mie and Rayleigh signals is carried out by two accumulation charge-coupled devices (ACCDs) with an array size of 16 x 16 pixels (image zone) and high quantum efficiency of 85% at 355 nm. For both channels, the electronic

charges of all 16 rows in the image zone are binned together to one row and stored in 25 rows of a memory zone, each row representing one range gate. Low readout noise is accomplished by directly accumulating charges in the memory zone within the CCD chip. In this way, the charges become sufficiently large that the noise contribution of the readout amplifier becomes negligible compared to the shot noise from the detected signal itself (Reitebuch et al., 2009). From the 25 range gates, three range gates are used for detecting the background light, signals resulting from a the electric voltage at the analogue-to-digital

converter (detection chain offset, DCO) and the internal reference signal, respectively. Two range gates act as buffers for the internal reference, so that atmospheric backscatter signals are collected in the remaining 20 range gates. Due to the transfer time from the image to the memory zone, the temporal resolution of one range gate is limited to 2.1 µs corresponding to a minimum range resolution of 315 m (height resolution of 296 m considering the 20°-off-nadir pointing of the instrument). The timing sequences of both ACCDs are flexibly programmable so that the vertical resolution within one wind profile can



be varied from 296 m to about 1.2 km, individually for the Mie and Rayleigh channel. The horizontal resolution of the A2D is determined by the acquisition time of the detection unit where the signals from 18 successive laser pulses are accumulated to so-called *measurements* (duration 0.4 s). Summation of the signals obtained from 35 measurements, i.e. 630 laser pulses, forms one *observation* (duration 14 s). Taking account of the time required for data read out and transfer (4 s), two subsequent observations are separated by 18 s.

For the satellite instrument, one observation consists of 30 accumulations (also referred to as measurements) of 19 shots, whereby data is continuously read out without gaps of 4 s. Hence, one observation takes 12 s. However, due to the much higher ground speed of Aeolus of about 7200 m·s$^{-1}$ compared to the Falcon aircraft (200 m·s$^{-1}$), the horizontal resolution of about 86.4 km is much coarser than for the A2D (3.6 km). In the course of the Aeolus wind retrieval, different accumulation lengths are possible depending on the signal strength in the Rayleigh and Mie channel, as explained in the next section.

**2.2 The Aeolus wind data product**

ALADIN on-board Aeolus is, like its airborne demonstrator, a direct-detection Doppler wind lidar which incorporates a frequency-stabilized UV laser and a dual-channel optical receiver to determine the Doppler shift from the broadband Rayleigh-Brillouin backscatter from molecules and the narrowband Mie backscatter from aerosols and cloud particles (ESA, 2008; Reitebuch, 2012). The major technical differences to the airborne instrument are the larger telescope diameter (1.5 m),

the larger slant angle (35°) and the free-path propagation of the internal reference signal, as explained above. An overview of the key instrument parameters of the two wind lidars is given in Table 1.

**Table 1.** Key instrument parameters of ALADIN and the ALADIN Airborne Demonstrator.

| Parameter | ALADIN | ALADIN Airborne Demonstrator (A2D) |
|---|---|---|
| Laser wavelength, repetition rate, pulse energy, linewidth | 354.8 nm<br>50.5 Hz<br>53…57 mJ (Nov. 2018)<br>30 MHz (FWHM) | 354.89 nm<br>50 Hz<br>60 mJ<br>50 MHz (FWHM) |
| Telescope diameter | 1.5 m | 0.2 m |
| LOS slant angle | 35° | 20° |
| Lidar principle | Direct-detection with double-edge and fringe imaging technique | Direct-detection with double-edge and fringe imaging technique |
| Receiver | Sequential Fabry-Pérot interferometers for molecular backscatter (Rayleigh channel) and Fizeau interferometer for particulate backscatter (Mie channel) | Sequential Fabry-Pérot interferometers for molecular backscatter (Rayleigh channel) and Fizeau interferometer for particulate backscatter (Mie channel) |
| Horizontal resolution | 86.4 km | 3.6 km |
| Vertical resolution | 250 m to 2000 m depending on range gate setting | 300 m to 1200 m depending on range gate setting |




The Aeolus Level 2B (L2B) product contains so-called horizontal line-of-sight (HLOS) winds for the Mie and Rayleigh channel. The L1B and L2B wind retrieval is described in detail in Algorithm Theoretical Basis Documents (Reitebuch et al., 2018; Tan et al., 2017). Thus, only a brief description is provided here. As a first step, Aeolus measurements (horizontal resolution of about 2.9 km corresponding to 0.4 s) are gathered together into groups where the length depends on the L2B
parameter settings. During the analysed period in November 2018 the group length was set to 30 Aeolus measurements corresponding to a horizontal extent of about 86.4 km. The measurement bins within the group are then classified into "clear" and "cloudy" bins using estimates of the scattering ratio. Before the wind retrieval is performed, the signals of the measurement bins from the same category are horizontally accumulated within the group. Separate wind retrievals are performed for both channels and for both categories, whereby only Rayleigh winds classified as clear and Mie winds
classified as cloudy are generally used for further analysis. In this manner, it is ensured that systematic errors introduced to the Rayleigh winds by contamination with particulate backscatter signals as well as low SNR of the Mie channel are avoided. Finally, to account for pressure and temperature effects in the Rayleigh wind retrieval (Dabas et al., 2008), a priori temperature and pressure information from ECMWF model results are interpolated along the Aeolus measurement track and used for correction. The utilized meteorological data is also included in an auxiliary data product (AUX_MET). It should be
mentioned that the Aeolus wind data obtained from the L2B product which is discussed here is in a preliminary state, inasmuch as biases related to known error sources are not corrected yet (Reitebuch et al., 2019; Rennie and Isaksen, 2019a). Such error sources will be elaborated in section 4.3.

In addition to the L2B wind product, Aeolus provides an L2C wind product which results from background assimilation of the Aeolus HLOS winds in the ECMWF operational prediction model. It contains vertical wind vector profiles ($u$ and $v$
components) and supplementary geophysical parameters.

## 3 Campaign overview, response calibration and wind observations

Only three months after the successful launch of Aeolus, the wind validation campaign (WindVal III) was conducted from the DLR airbase in Oberpfaffenhofen, Germany in the in the time frame from 5 November to 5 December 2018. The campaign represented a continuation of the previous field experiments WindVal I in 2015 (Marksteiner et al., 2018) and
WindVal II (NAWDEX) in 2016 (Schäfler et al., 2018; Lux et al., 2018) which were performed from Keflavík, Iceland. The previous campaigns aimed at the pre-launch validation of the Aeolus mission by exploiting the high degree of commonality of the A2D with the satellite instrument to test its measurement principle and to refine its wind retrieval algorithms based on real atmospheric measurements. With Aeolus operating in space, the objectives of WindVal III went beyond those of the preceding campaigns. For the first time, collocated wind measurements of ALADIN and its airborne demonstrator could be
performed offering the possibility to compare the performance of both instruments under various atmospheric conditions. In addition to the collocated wind observations shortly after launch, one goal of the WindVal III campaign was to rehearse the validation activities to be performed after the commissioning phase of the Aeolus mission. This included, first and foremost,





the planning of the flights along the satellite measurement track which required a thorough consideration of the weather conditions along the swath within the reach of the DLR Falcon research aircraft, air traffic control limitations as well as the

satellite status and operating hours of the Oberpfaffenhofen airport. For the purpose of high wind data coverage of Aeolus, target areas without high- or mid-level clouds were generally preferred for the underflights. Ideally, the flights included sections with cloud-free conditions, as this allowed for strong ground return signals which could be exploited for reducing potential wind biases by means of zero wind correction (Marksteiner, 2013; Lux et al., 2018).

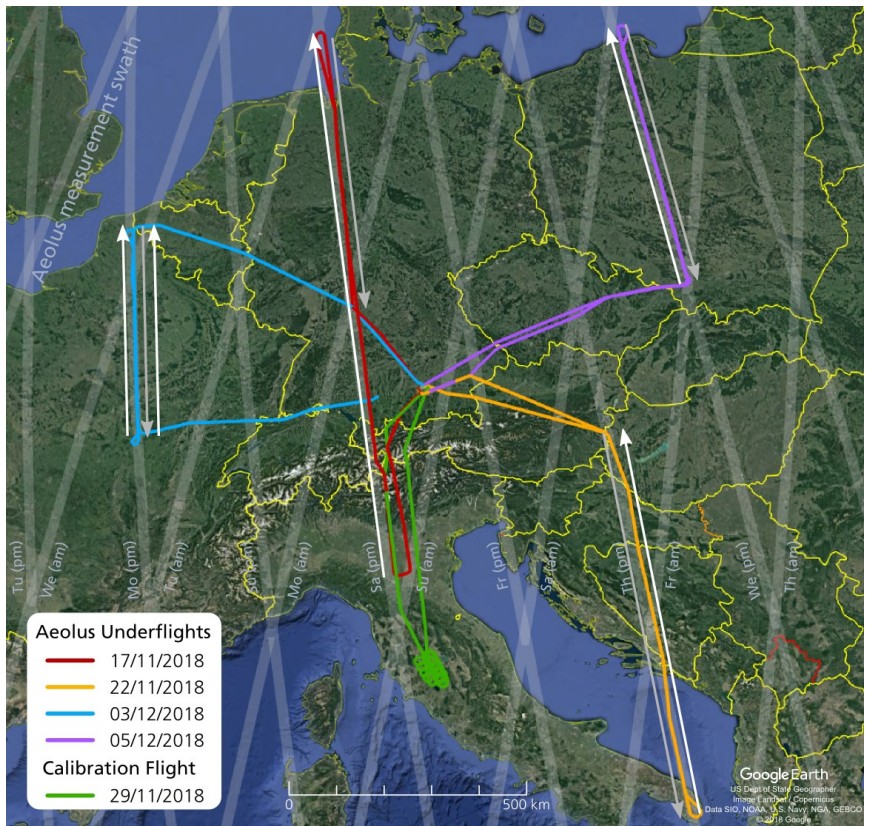

**Figure 2.** Flight tracks of the Falcon aircraft during the WindVal III campaign from 17 November to 5 December 2018 (background

image: © 2018 Google). Each colour represents a single flight. The Aeolus measurement swath is shown in grey colour. The arrows indicate the Falcon flight direction along the swath on the different legs in (white arrows) and against (grey arrows) the satellite direction which was always from south to north during the probed evening satellite tracks. The A2D was not operable during the flight on 17 November 2018.

In the framework of the WindVal III campaign in autumn 2018, six flights were conducted including a test flight and a calibration flight. The corresponding flight tracks of the DLR Falcon aircraft are shown in Fig. 2 together with the swaths of the Aeolus satellite for one week. A total of 22 flight hours were carried out including the test flight performed a few hours before the first underflight on 17 November 2018. Adding up the lengths of the satellite swaths covered by the aircraft during the four underflights, the overall track length for which wind data was acquired for validation purposes is nearly

3000 km. The first underflight was also the longest flight along the Aeolus track (1155 km) covering the measurement swath





from Northern Italy up to the North Frisian Islands. While the 2-µm DWL was operating without limitations during the entire campaign, the A2D was not operational during the first flight due to technical issues, so that A2D wind data is only available from the three other underflights. The data obtained along the Aeolus track is subdivided into seven wind scenes corresponding to the flight legs indicated by arrows in Fig. 2. An overview of these scenes including the number of A2D

observations are presented in Table 2 together with the geolocations of the start and end points of the respective flight legs. The number of Aeolus observations for each scene is provided as well.

**Table 2.** Overview of the research flights of the Falcon aircraft in the frame of the WindVal III campaign and the wind scenes performed with the A2D along the Aeolus measurement track. The A2D was not operable during the flight on 17 November 2018.

| Flight # | Date | Flight period (UTC) | Measurement period (UTC) | Number of A2D observations | Geolocation of DLR Falcon on Aeolus measurement track (start/stop) | | Number of Aeolus observations |
|---|---|---|---|---|---|---|---|
| 1 | 17/11/2018 | 15:14 – 19:14 | A2D inoperable | No data | 44.7°N, 10.6°E | 54.9°N, 7.8°E | 12 |
| **2** | **22/11/2018** | **14:29 – 17:56** | 15:11 – 15:48 | 122 | 46.7°N, 16.8°E | 42.3°N, 17.7°E | 7 |
| | | | **16:13 – 17:15** | **176** | **40.5°N, 18.1°E** | **47.2°N, 16.5°E** | **9** |
| 3 | 29/11/2018 | 09:56 – 14:00 | Calibration flight | | | | |
| 4 | 03/12/2018 | 15:48 – 19:31 | 16:48 – 17:13 | 82 | 47.8°N, 3.5°E | 50.5°N, 2.8°E | 4 |
| | | | 17:22 – 17:48 | 87 | 50.1°N, 2.9°E | 46.8°N, 3.7°E | 4 |
| | | | 17:53 – 18:29 | 117 | 47.1°E, 3.6°E | 50.6°N, 2.7°E | 5 |
| 5 | 05/12/2018 | 14:56 – 18:22 | 15:53 – 16:45 | 173 | 50.3°N, 18.9°E | 54.9°N, 17.6°E | 7 |
| | | | 16:55 – 17:18 | 78 | 54.0°N, 17.9°E | 50.8°N, 18.8°E | 4 |

## 3.1 Response calibrations

The flight on 29 November 2018 was dedicated to the calibration of the A2D which is a prerequisite for the wind retrieval, since the relationship between the Doppler frequency shift of the backscattered light, i.e. the wind speed, and the response of the two spectrometers has to be known for the wind retrieval. In particular, the frequency dependence of the Rayleigh response has to be determined for different altitudes, since the spectral shape of the Rayleigh-Brillouin backscatter signal influencing the transmission through the two FPI filters significantly depends on temperature and pressure of the sampled

atmospheric volume (Witschas et al., 2010; Dabas et al., 2008) and thus varies along the laser beam path. Calibration of the Rayleigh and Mie channel involves a frequency scan of the laser transmitter over 1.4 GHz (±125 m·s$^{-1}$) to simulate well-defined Doppler shifts of the atmospheric backscatter signal within the limits of the laser frequency stability. During this procedure the contribution of (real) wind related to molecular or particular motion along the instruments' LOS is virtually

eliminated by flying curves at a roll angle of the Falcon aircraft of 20°, hence resulting in approximate nadir pointing of the instrument and, in case of negligible vertical wind, vanishing LOS wind speed. In the course of one frequency scan which takes about 24 minutes, unknown contributions to the Rayleigh and Mie response such as temperature variations of the spectrometers or frequency fluctuations of the laser transmitter have to be minimized, as they can introduce systematic errors or increase the random error of the derived wind speed. Above all, cloud-free conditions are necessary to avoid Mie





backscatter signals which affect the backscatter spectrum, and thus the Rayleigh response in the respective range gates. Furthermore, ground visibility is required for calibrating the Mie channel. Additional information on the A2D calibration procedure and how it compares to the satellite mission are comprehensively described in Marksteiner et al. (2018), while details on the calibration and wind retrieval of Aeolus can be found in Tan et al. (2016) and Reitebuch et al. (2018).

The region between Rome and Florence with clear atmosphere and nearly zero vertical wind was chosen for the WindVal III

calibration flight on 29 November 2018. The green track in Fig. 2 shows the characteristic circular flight pattern in Northern Italy which follows from the 20° roll angle of the aircraft for the purpose of nadir pointing of the A2D. In the period from 10:48 UTC to 12:51 UTC, four response calibrations, i.e. laser frequency scans, were performed to obtain four sets of calibration parameters. Based on several quality criteria which were identified during previous campaigns and are mostly related to instrument housekeeping data (Marksteiner et al., 2018), one of the four calibration sets was selected for Rayleigh

and Mie wind retrieval, respectively. The chosen Rayleigh calibration was especially characterized by a high pointing stability of the laser transmitter which is of high importance for assuring a low random error of the Rayleigh channel, as even small variations in the incidence angle on the Rayleigh FPIs by a few μrad largely influence the Rayleigh response, potentially leading to wind errors of several m·s$^{-1}$ (DLR, 2016). Moreover, the selected calibration showed the smallest residuals of the fifth-order polynomial fit applied to Rayleigh response curve, thus ensuring the lowest random wind error

which may result from discrepancies between the calibration fit function and the actual frequency dependence of the spectrometer response. For the Mie channel, the four calibration results were very consistent which can be traced back to the integration of the fiber scrambler that considerably reduced the speckle noise of the internal reference signal (Lux et al., 2019). Hence, since there were no additional arguments in favour or against a certain calibration, the one with the lowest temperature variability of the Fizeau interferometer was selected for the Mie wind retrieval.

**3.2 A2D wind results from the underflight on 22 November 2018**

First collocated wind observations of the A2D and Aeolus were performed on 22 November when the Falcon flew along the satellite swath from Lecce in South Italy (40.5°N, 18.1°E) to the Austrian-Hungarian border (47.2°N, 16.5°E) (see Fig. 2). Aeolus covered this track between 16:34:14 UTC and 16:36:02 UTC, while it took the Falcon more than one hour from 16:13 UTC to 17:15 UTC to travel the distance of about 790 km. Cloud-free conditions and strong winds prevailed in the

southern part of the leg, while mid-level clouds and weak winds occurred for the northern part in accordance with the weather prediction used for flight planning.

During the underflight, the A2D performed 176 wind observations while wind data from nine observations was acquired by Aeolus (see Table 2). The A2D wind scene was deliberately interrupted by a so-called MOUSR (Mie Out of Useful Spectral Range) measurement between 16:45 UTC and 16:54 UTC. This mode is used to detect the Rayleigh background signal

distribution on the Mie channel which is important for quantifying the broadband molecular return signal transmitted through the Fizeau interferometer. For this purpose, the laser frequency was tuned away by 1.05 GHz from the Rayleigh filter cross point and Mie channel centre which defines the set frequency during the wind scenes. As a result, the laser frequency of the



emitted pulses was outside of the useful spectral range of the Mie spectrometer, so that the fringe was not imaged onto the Mie ACCD and only the broadband Rayleigh signal was detected on the Mie channel. The range-dependent intensity levels
per pixel were subsequently subtracted from the measured Mie raw signal in order to avoid systematic errors in the determination of the fringe centroid position and, in turn, in the Mie winds.

The Rayleigh and Mie signal intensities per observation are shown in Fig. 3. The raw signals were first corrected for the solar background and the DCO which are collected in two dedicated range gates, as explained above. Range correction (normalization to 1 km) was then applied taking into account that the intensity decreases as the inverse square of the distance
between the scatterer and the detector. Finally, the integration times set for each range gate were considered for normalising the signal intensities per range gate to a bin size of 296 m (2.1 µs integration time). While the intensity profile for the Rayleigh channel essentially follows the vertical distribution of the atmospheric molecule density, the Mie intensity profile displays the vertical distribution of atmospheric cloud and aerosol layers along the flight track. High Rayleigh signal intensities can be attributed to cloud layers at different altitudes along the flight track which also manifest in increased Mie
signal intensities.

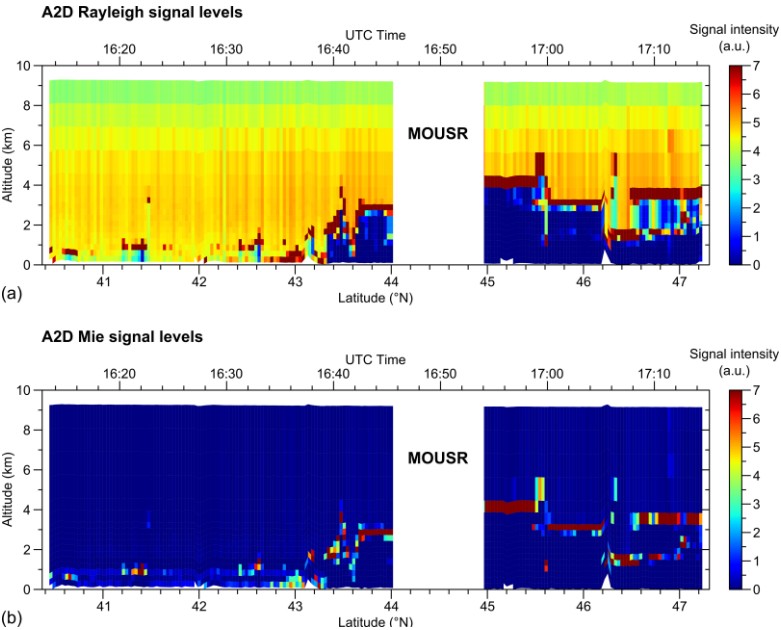

**Figure 3.** Background and DCO-corrected signal levels from (a) the A2D Rayleigh channel and (b) the Mie channel measured during the underflight on 22 November 2018 between 16:14 UTC and 17:14 UTC along the Aeolus measurement track. Between 16:45 UTC and
16:54 UTC the A2D was operated in a different mode (MOUSR) aiming at the detection of the Rayleigh background signal on the Mie channel.

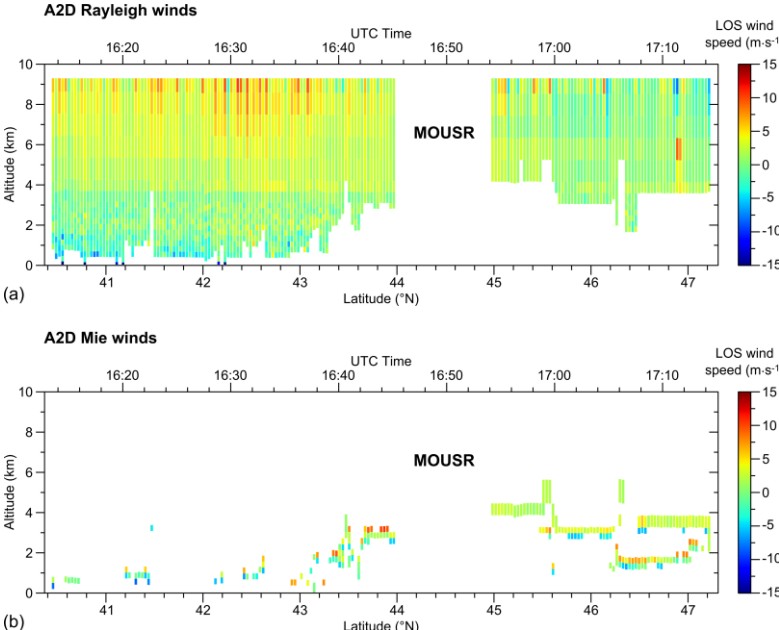

**Figure 4.** LOS wind profiles (positive if winds are blowing away from the instrument) measured during the underflight on 22 November
2018 between 16:13 UTC and 17:15 UTC along the Aeolus measurement track (white arrow in Fig. 2) using (a) the A2D Rayleigh and (b)
the Mie channel. White colour represents missing or invalid data due to low signal, e.g. below dense clouds. The data gap between 16:45
UTC and 16:54 UTC is due to an interruption of the wind measurement during a different operation mode (MOUSR) of the A2D
instrument aiming at the detection of the Rayleigh background signals on the Mie channel.

Figure 4 shows the processed LOS Rayleigh and Mie winds plotted versus latitude (and time) and altitude for the period of

the Aeolus underflight on 22 November 2018. During the first section of the flight, cloud-free conditions led to nearly

complete data coverage of the Rayleigh channel from ground up to 9 km altitude. In the second half of the flight, dense mid-

level clouds limited the extension of the Rayleigh wind profiles to above 4 km to 5 km. The data gap in between is due to the

MOUSR procedure mentioned above.

The range gate settings were identical for the Rayleigh and Mie channel and chosen to sample the lowermost 3.5 km of the

troposphere with the highest possible vertical resolution. Therefore, the integration time of the ACCD was set to 8.4 μs in the

range gates 7 to 10 (9.3 km to 4.5 km), 4.2 μs in the range gates 11 and 12 (4.5 km to 3.5 km) and 2.1 μs in all the remaining

range gates towards the ground, corresponding to a height resolution of 1184 m, 592 m and 296 m, respectively. LOS wind

speeds of up to 15 m·s$^{-1}$ were measured with the Rayleigh channel at altitudes between 8 km and 9 km. Note that positive

wind speeds are obtained when the A2D LOS unit vector points along the direction of the horizontal wind vector, i.e. the

wind is blowing away from the instrument. This definition is in contrast to previous campaigns where winds blowing

towards the instrument were defined positive in accordance with a positive Doppler frequency shift. It was inverted in order

to follow the sign convention of Aeolus, thus allowing for a better comparison different wind data sets. In the shown case,

northwesterly winds were present around the Adriatic Sea with horizontal wind speeds up to 50 m·s$^{-1}$ at 9 km altitude

according to the ECMWF model. However, as the A2D was pointing towards the northeast along the Aeolus track, the



projection of the horizontal wind vector onto the instrument's LOS was small, resulting in low measured wind speeds with positive sign. At lower altitudes, the wind direction was opposite, so that the wind was blowing towards the instrument, leading to slightly negative wind speeds.

In contrast to the Rayleigh channel, the Mie data coverage is rather poor owing to the sparse cloud cover and low aerosol load during the flight. Wind data is mainly obtained from the cloud tops along the track. Due to the high optical density of
the clouds, the laser was strongly attenuated, thus preventing sufficient backscatter signal and valid Mie wind data over multiple range gates across the clouds. As a result, valid Mie wind data is often obtained for only one bin per profile or, in case data from a subjacent range gate passes quality control, the wind data shows a large systematic error. This is likely due to the skewness of the Mie fringe on the ACCD which influences the determination of the centroid position depending on the position of the cloud within the range gates. The same characteristics were observed for the other two Aeolus underflights,
so that the number of valid and good quality Mie wind data is very low compared to the Rayleigh channel. The scarce coverage of the Mie data and the high number of outliers due to the Mie fringe skewness in combination with the presence of thick clouds prevented a meaningful comparison with the Aeolus data which showed similarly poor Mie data coverage for the same reasons. Thus, the further analysis of the A2D and Aeolus wind data is restricted to the Rayleigh channel.

### 3.3 Aeolus wind results from the underflight on 22 November 2018

When the Falcon aircraft was located at 42.8°N, 17.7°E at 16:34:56 UTC after about one third of the common leg from Lecce to the Austrian-Hungarian border, Aeolus was just passing by, measuring winds in the same atmospheric volume along its path. 66 seconds later, the satellite finished the common leg, while the Falcon arrived at the northern end of the track at 17:15 UTC, resulting in a maximum temporal distance between the wind data acquisitions of the airborne and satellite instruments of about 39 minutes. The wind data obtained with the Aeolus Rayleigh channel is depicted in Fig. 5.
The profiles span the range from the ground to the lower stratosphere (21 km) with vertical resolution of 0.25 km in the lowermost range gates up to 2 km altitude. In the range between 2 km and 13 km altitude the bin thickness is 1 km, while it is 2 km in the region above. Hence, a maximum of 15 range bins lie within the sampled altitude range of the A2D below 10 km. The selected range gate setting of Aeolus ensured accurate ground detection with the highest possible vertical resolution which was crucial for determining potential wind biases during the commissioning phase of the mission. The data
plotted in Fig. 5 are the wind speeds measured along the satellite's LOS which is asterisked (LOS*) in the following in order to avoid confusion with the A2D LOS. Due to the larger off-nadir angle of 37° relative to the normal direction at the measurement swath (considering the Earth curvature) compared to the airborne demonstrator (20°), the projection of the horizontal wind vector onto the satellite LOS is generally larger and the measured wind speeds are thus higher.

The HLOS wind speed $v^*_{\text{Aeolus, HLOS}}$ included in the L2B wind data product of Aeolus can be converted to the LOS* wind
speed $v^*_{\text{Aeolus, LOS}}$ via the off-nadir angle $\Theta_{\text{Aeolus}}$:

$$v^*_{\text{Aeolus, LOS}} = v^*_{\text{Aeolus, HLOS}} \cdot \sin(\theta_{\text{Aeolus}}).$$ (1)





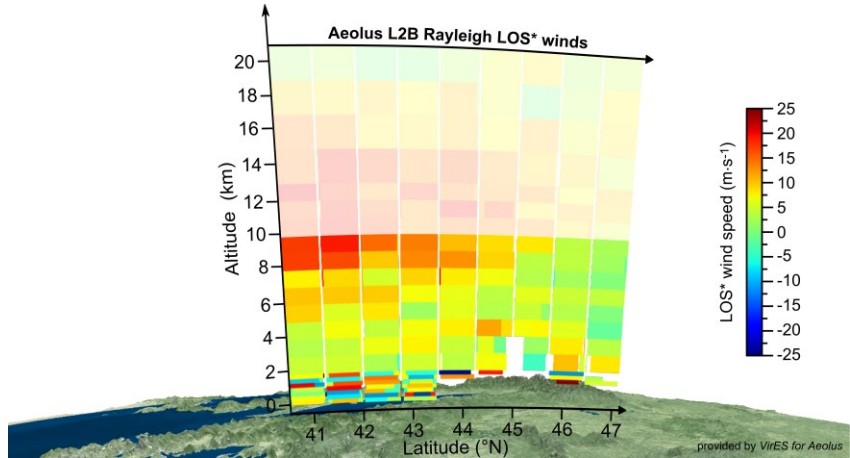

**Figure 5.** Aeolus L2B LOS* Rayleigh winds (positive if winds are blowing away from the instrument) measured during the underflight on 22 November 2018 between 40.6°N and 47.2°N. Only winds with an estimated wind error of less than 12 m·s$^{-1}$ are shown. Winds at altitudes above 10 km are outside of the measurement range of the A2D and therefore shown greyed out. The figure was created based on a screenshot from the Aeolus visualization tool *VirES for Aeolus* (*https://aeolus.services/*).

The L2B product also contains the Rayleigh estimated wind error which is derived from the signal-to-noise level and the pressure and temperature sensitivity of the Rayleigh channel responses (Tan et al., 2017). Bins for which the estimated error is larger than 12 m·s$^{-1}$ are omitted in the diagram in Fig. 5. This leads to data gaps in the lower troposphere in the northern part of the common leg where dense low-level clouds strongly attenuated the laser beam and the backscattered signal from below the clouds, as also observed for the A2D. In accordance with the weather forecast, Aeolus measured strong winds in the southern part of the leg between 8 and 10 km, reaching LOS* wind speeds of up to 25 m·s$^{-1}$, whereas weaker winds were observed towards the north. Before comparing the A2D and Aeolus wind results from the selected underflight as well as the entire campaign, the quality of the A2D data during WindVal III will be discussed in the following.

## 3.4 Assessment of the A2D performance by comparison with the 2-µm DWL

The accuracy and precision of the A2D Rayleigh wind results were evaluated by comparing them to the wind data obtained from the coherent 2-µm DWL which was operated in parallel on the same aircraft and which is characterized by high accuracy of the horizontal wind speed of about 0.1 m·s$^{-1}$ and precision of better than 1 m·s$^{-1}$ (Weissmann et al., 2005; Chouza et al., 2016; Witschas et al., 2017). For this purpose, the three-dimensional wind vectors measured with the 2-µm DWL were projected onto the A2D LOS axis. Moreover, the 2-µm measurement grid was adapted to that of the A2D by means of a weighted aerial interpolation algorithm, as introduced in Marksteiner et al. (2011). The latter was, in a similar way, also utilized for the comparison of the A2D and Aeolus data and will be described in the next section. In analogy to the results presented for selected flights of the WindVal II campaign in 2016 (Lux et al., 2018), a statistical comparison was performed yielding the systematic and random error of the A2D Rayleigh winds for all underflights of the WindVal III campaign. The corresponding scatterplot is depicted in Fig. 6(a) together with the results from the previous campaign WindVal II, while the respective statistical parameters are provided in Table 3.



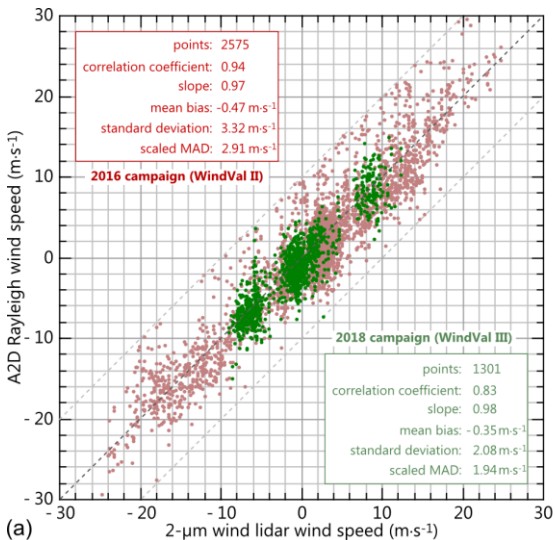
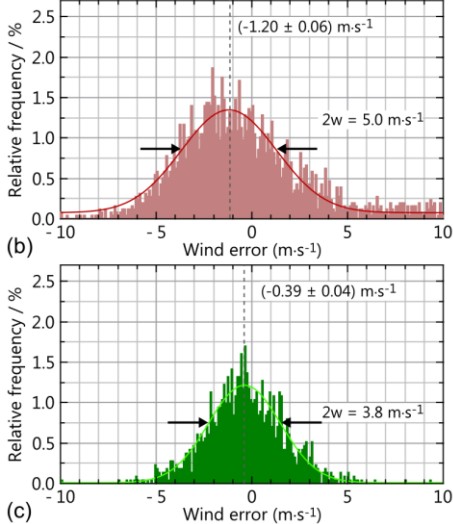

**Figure 6.** (a) Scatterplots comparing the A2D Rayleigh LOS winds with the 2-µm DWL winds for all Aeolus underflight legs from the WindVal III campaign in 2018 (green) and from all flights of the WindVal II campaign in 2016 (red). The corresponding probability density functions for the wind differences (A2D–2 µm), i.e. the A2D wind error, are shown in panels (b) and (c) for the two campaigns, respectively. The solid lines represent Gaussian fits with the given centres and $e^{-1/2}$-widths $2w$.

In addition to the parameters provided in the insets of Fig. 6(a), the table also includes the slopes $A$ and intercepts $B$ from non-weighted linear fits $v_y = A \cdot v_x + B$ applied to the two scatterplots. Here, $x$ and $y$ represent the values plotted on the abscissa and ordinate, respectively. The standard error of the slope given in the table was calculated according to

$$s_A = \sqrt{\frac{\frac{1}{n-2}\sum_{i=1}^{n}\varepsilon_i^2}{\sum_{i=1}^{n}(v_{x,i}-\overline{v_x})^2}}, \text{ with} \tag{2a}$$

$$\varepsilon_i = v_{y,i} - \left(A \cdot v_{x,i} + B\right) \tag{2b}$$

being the residuals of the linear regression.

**Table 3.** Results of the statistical comparison between the A2D Rayleigh channel and the 2-µm DWL wind data for all flights performed during the WindVal II campaign in 2016 and the WindVal III campaign in 2018. See corresponding scatterplots in Fig. 6. The values are given as wind speeds measured along the A2D LOS.

| Statistical parameter | WindVal II | WindVal III |
|---|---|---|
| Number of compared bins | 2575 | 1301 |
| Correlation coefficient $r$ | 0.94 | 0.83 |
| Slope $A$ | $0.97 \pm 0.02$ | $0.98 \pm 0.02$ |
| Intercept $B$ | -0.4 m·s⁻¹ | -0.4 m·s⁻¹ |
| Mean bias | -0.5 m·s⁻¹ | -0.4 m·s⁻¹ |
| Standard deviation | 3.3 m·s⁻¹ | 2.1 m·s⁻¹ |
| Scaled MAD | 2.9 m·s⁻¹ | 1.9 m·s⁻¹ |





During the WindVal II campaign twelve research flights were performed with a primary focus on the sampling of high wind speeds and gradients related to the North Atlantic jet stream. Therefore, the number of compared wind results and the wind
speed range are considerable larger compared to the WindVal III campaign where generally weaker winds were encountered during the four satellite underflights in Central Europe. Nevertheless, more than half as many A2D Rayleigh winds entered the statistical comparison with the 2-µm DWL winds, as the WindVal III flights were preferentially conducted in cloud-free regions for the purpose of large data overlap with Aeolus. It should be noted here that the 2-µm DWL is very sensitive to weak backscatter return from clouds and aerosols due to its small-bandwidth coherent detection principle, so that 2 µm DWL
winds are even available for low scattering ratios (< 1.1), where insignificant Mie contamination of the A2D Rayleigh channel can be expected.

The WindVal III flight planning aimed to reduce the probability for heterogeneous cloud conditions which, in turn, increased the representativity of the scan-retrieved volume winds obtained from the 2-µm DWL to the A2D LOS winds. Furthermore, the risk for large systematic errors of the Rayleigh channel, e.g. introduced by cirrus clouds affecting the transmit-receive co-
alignment feedback loop, was minimized. Consequently, the scatterplot for WindVal III in Fig. 6(a) features much less outliers compared to that of WindVal II. The more homogeneous atmospheric conditions in combination with the implementation of the fiber scrambler to diminish the internal reference frequency noise result in a significant reduction of the random error by more than 30% to less than 2 m·s$^{-1}$, while the mean bias of -0.4 m·s$^{-1}$ is comparable to the previous campaign (-0.5 m·s$^{-1}$).
In addition to the standard deviation, the median absolute deviation (MAD) was determined for quantifying the random error of the A2D wind speed measurements. It is defined as the median of the absolute variations of the measured wind speeds from the median of the wind speed differences:

$$\mathrm{MAD} = \mathrm{median}\big[\big|(v_{\mathrm{A2D},i} - v_{\mathrm{ECMWF},i}) - \mathrm{median}(v_{\mathrm{A2D},i} - v_{\mathrm{ECMWF},i})\big|\big]. \tag{3}$$

Compared to the standard deviation, the MAD is more resilient to outliers and thus a more robust measure of the variability
of the measured wind speeds. In case that the random wind error is normally distributed, the MAD value, multiplied by 1.4826 (scaled MAD), is identical to the standard deviation. The larger number of outliers in the scatterplot for the WindVal II campaign manifests in a larger discrepancy between the standard deviation (3.3 m·s$^{-1}$) and the scaled MAD value (2.9 m·s$^{-1}$) compared to WindVal III (2.1 m·s$^{-1}$, 1.9 m·s$^{-1}$). The random error can also be approximated from probability density functions (PDFs) illustrating the frequency distribution of the wind speed differences $v_{\mathrm{A2D}} - v_{2\mu\mathrm{m}}$, i.e. the wind error
(Fig. 6(b,c)). Since the wind error is not perfectly Gaussian-distributed for both campaigns, there is a deviation between the mean bias values and the center of the Gaussian fits. For the same reason, the width of the fits is narrower than twice the standard deviations which also consider the outliers.





## 4 Comparison of A2D and Aeolus wind data

For adequate comparison of the A2D wind profiles with the Aeolus wind data, two major aspects have to be considered.
First, the different horizontal and vertical resolutions of the two instruments necessitate an adaptation of the A2D measurement grid to that of Aeolus. Second, the different viewing geometries of the wind lidars need to be taken into account. Since both instruments measure only one component of the horizontal wind vector along the respective LOS, information on the wind direction is required in order to determine the wind speed difference of the A2D and Aeolus resulting from the different LOS directions.

### 4.1 Adaptation of the measurement grid

Due to the fact that the horizontal resolution of Aeolus is much coarser than that of the A2D (see Table 1), interpolation of the A2D wind measurements onto the Aeolus measurement grid is required. In the framework of previous A2D campaigns, an aerial weighted averaging algorithm (Marksteiner et al., 2011; Marksteiner, 2013) was developed to compare the A2D wind results with the data obtained from the 2-μm reference wind lidar (Lux et al., 2018). The grid adaptation procedure
used in this study is based on that algorithm. Each valid A2D range bin covering an Aeolus range bin is allocated horizontal and vertical weights depending on the size of the contribution of the respective A2D bin to the total area of the Aeolus bin, as illustrated in Fig. 7.

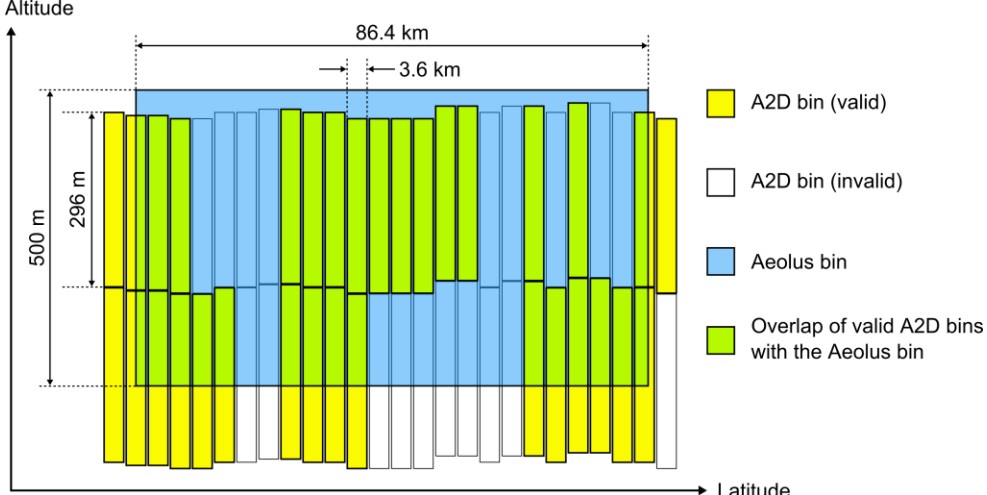

**Figure 7.** Schematic illustrating the different horizontal and vertical resolutions of Aeolus (blue bin) and the A2D (yellow bins) for typical range gate settings of the two instruments (Aeolus: 500 m vertical resolution, A2D: 296 m vertical resolution). White bins indicate invalid A2D observations while the green area represents the overlap of valid A2D bins with the Aeolus bin. For the aerial weighted averaging algorithm the contributions of each valid A2D wind value to the wind value allocated to the composite bin are weighted by the overlap of the respective A2D bins with the regarded Aeolus bin. The ratio of the green to the blue area is defined as coverage ratio and used as a
quality control parameter (see section 4.4).

Hence, for each Aeolus range bin a weighted average from the A2D contributions can be calculated. Moreover, the coverage ratio which determines the coverage of an Aeolus range bin by valid A2D bins is calculated as a measure of the representativity of A2D winds within an Aeolus range bin. Especially in regions with strong wind shear within the area of an Aeolus range bin and sparse coverage, large representativity errors are possible, e.g. when the A2D bins only cover the area

of an Aeolus bin where high wind speeds reside, while lower wind speeds within the Aeolus bin are not covered. The influence of the coverage ratio on the statistics of the wind comparisons is discussed in section 4.4. Additionally, the mean distance of the horizontal centers of the A2D bins covering an Aeolus bin to the latter's bin center was defined as a second adjustable parameter which potentially influences the outcome of the statistical comparison.

### 4.2 Consideration of the different viewing geometries

A visual comparison of the A2D Rayleigh winds in Fig. 4(a) measured during the underflight on 22 November 2018 with the corresponding Aeolus L2B Rayleigh wind curtain shown in Fig. 5 reveals large discrepancies. This is due to the fact that the viewing angles of the two instruments differ from each other. First of all, the off-nadir angles are different, as stated above ($\Theta_{Aeolus} = 37°$, $\Theta_{A2D} = 20°$). Additionally, depending on the wind speed and direction along the flight track, the heading angle of the aircraft deviates from the course angle (side-slip), resulting in a varying azimuth angle of the A2D. The situation is

illustrated in Fig. 8 showing the flight track of the aircraft along the satellite measurement swath together with the respective horizontal pointing directions of the A2D and Aeolus at a selected position on the track. While the azimuth angle of the A2D was around 68°, it was 80° for Aeolus. As a result, the two instruments measured different components of the horizontal wind vector projected onto the respective LOS vectors. In order to convert the A2D LOS winds to A2D LOS* winds, i.e. A2D winds that would have been measured if the airborne demonstrator was pointing along the same direction as the

satellite instrument, both the off-nadir and the azimuth angle need to be considered. In a first step, the A2D LOS winds are converted to A2D HLOS winds in analogy to Eq. (1). Then, the real wind speed difference Δ which results from the different azimuth angles of the two instruments has to be determined and added to the actual wind speed measured by the A2D:

$$v^*_{A2D} = v_{A2D} + \Delta. \tag{4}$$

The determination of Δ requires an additional source of information. For this purpose, model wind data from the ECMWF

which is included in the Aeolus L2C data product was utilized. Knowledge of the zonal ($u$) and meridional ($v$) wind component allows calculating the wind speed difference introduced by the different azimuth angles of the A2D ($\varphi_{A2D}$) and Aeolus ($\varphi_{Aeolus}$) as follows:

$$\Delta = [\sin(\varphi_{A2D}) - \sin(\varphi_{Aeolus})] \cdot u + [\cos(\varphi_{A2D}) - \cos(\varphi_{Aeolus})] \cdot v. \tag{5}$$

Using the above mentioned azimuth angles for the two instruments ($\varphi_{A2D} = 68°$ and $\varphi_{Aeolus} = 80°$), the wind speed difference

Δ can be larger than 5 m·s⁻¹ for typical zonal and meridional wind speeds of ±30 m·s⁻¹, as shown in the inset of Fig. 8. Only in case that the horizontal wind vector bisects the angle between the A2D and the Aeolus azimuth angle, that is for a wind direction of ($\varphi_{A2D} + \varphi_{A2D}$)/2 = 74° in the present example, the wind speed difference vanishes (Δ = 0). On the contrary, the deviation is maximum when the horizontal wind vector is perpendicular to the case mentioned before (-16° or 164°).



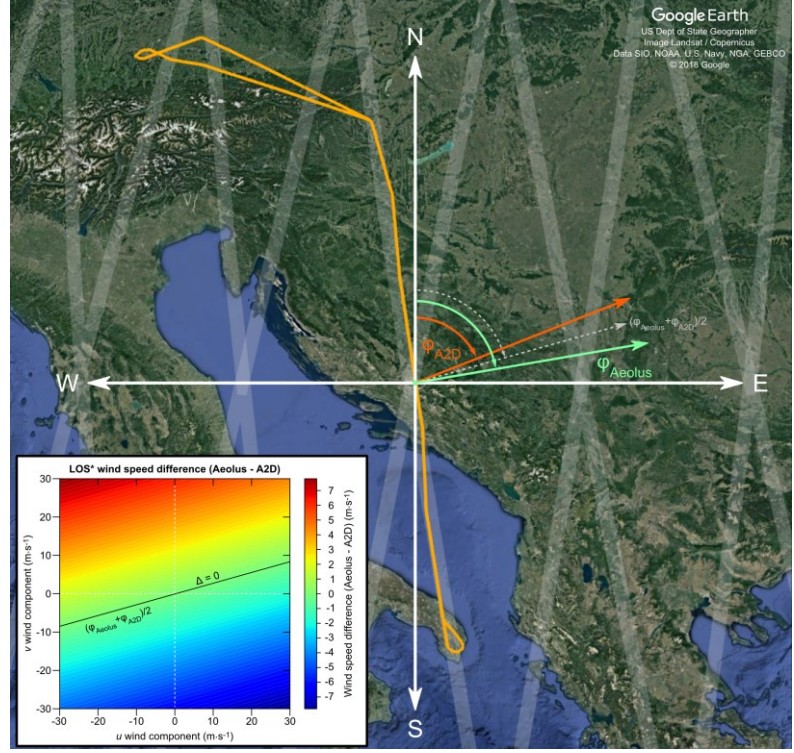

**Figure 8.** Diagram illustrating the different azimuth angles of Aeolus (green) and the A2D (orange) at the example of the underflight on 22 November 2018 indicated by the orange flight track (background image: © 2018 Google). The inset depicts the LOS* wind speed difference in dependence on the zonal ($u$) and meridional ($v$) wind component for azimuth angles of $\varphi_{A2D} = 68°$ and $\varphi_{Aeolus} = 80°$. In case the azimuth angle of the wind vector meets the condition $\varphi_{wind} = (\varphi_{Aeolus}/\varphi_{A2D})/2$, i.e. the wind direction is 74° (dashed grey arrow), the LOS* wind speed difference $\Delta$ is zero.

Since the Aeolus azimuth angle is generally around 80° in mid-latitudes on ascending orbits and the A2D azimuth angle is around 68° when flying on ascending satellite tracks, the plot is representative for all underflights of the WindVal III campaign. Thus, it can generally be stated, that the meridional wind component predominantly influences the wind speed difference $\Delta$. In summary, it can be stated that the azimuth correction is essential for accurate comparison of the A2D and Aeolus winds.

Despite the high accuracy and precision of the 2-μm DWL, the model data was utilized for the azimuth correction because of its full coverage. In contrast, the coherent detection 2-μm DWL data exhibits gaps in clear air regions, thus preventing the correction of many wind results obtained with the A2D Rayleigh channel. It should be mentioned that, in principle, the adapted A2D wind results can be potentially impacted by a model error (Schäfler et al., 2019). However, the comparison of the ECMWF model winds, averaged onto the Aeolus grid, with the 2-μm DWL wind data showed excellent agreement (bias below 0.1 m·s⁻¹, random error ≈ 2 m·s⁻¹) without any significant outliers over the entire campaign (Witschas et al., 2019). Finally, the azimuth-corrected A2D HLOS, i.e. the A2D HLOS*, wind speeds are multiplied by the factor sin(37°) ≈ 0.60 (see Eq. (1)) to obtain the A2D LOS* winds. The resulting wind curtains are depicted in Fig. 9.

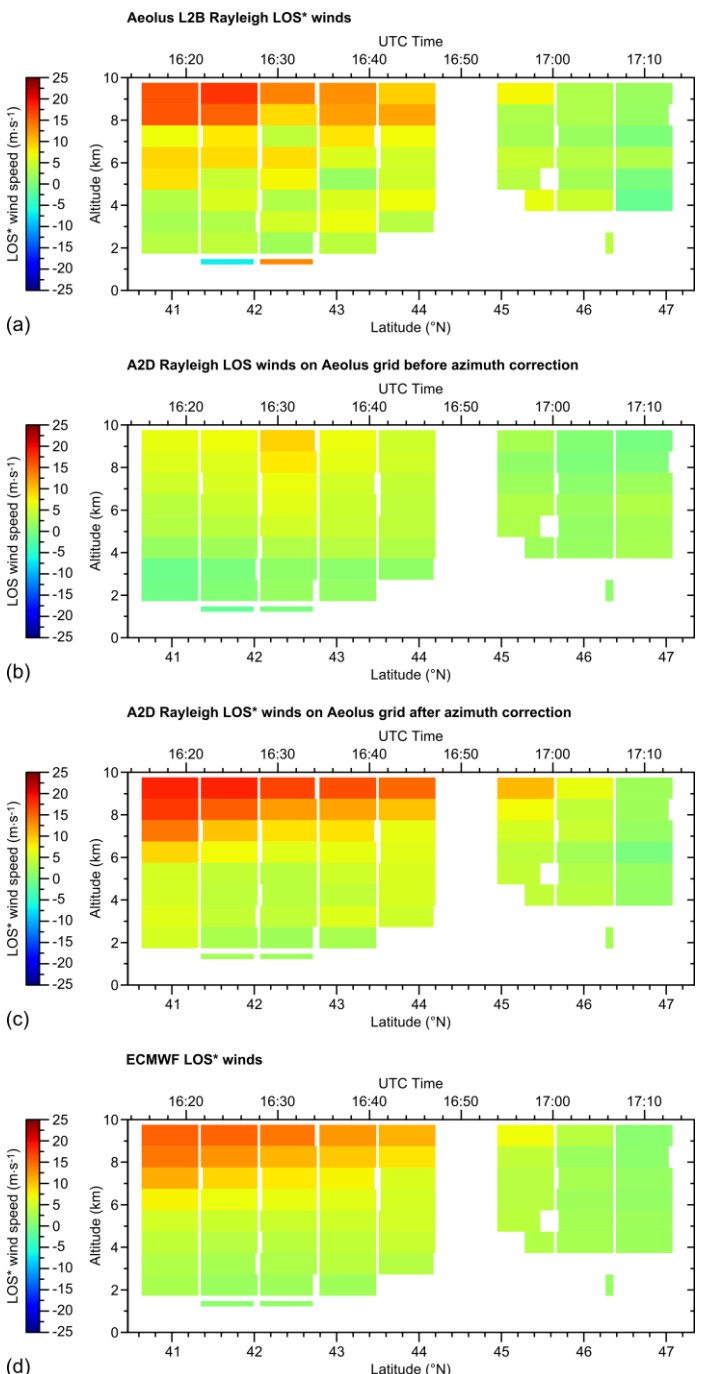

**Figure 9.** LOS* wind profiles obtained during the underflight on 22 November 2018 between 40.5°N and 47.2°N: (a) Aeolus L2B Rayleigh winds, (b) A2D Rayleigh winds averaged onto the Aeolus measurement grid and for an off-nadir angle of 37°, but without azimuth correction, (c) A2D Rayleigh winds with azimuth correction and (d) ECMWF model winds. White colour represents missing or invalid data of one of the two instruments, e.g. below dense clouds. Only Aeolus Rayleigh LOS* winds with an estimated error below 4.8 m·s$^{-1}$ were considered valid.

The figure shows the Aeolus L2B Rayleigh winds; the A2D Rayleigh winds averaged onto the Aeolus measurement grid and
for an off-nadir angle of 37°, but without azimuth correction; the A2D Rayleigh winds with azimuth correction (A2D LOS*
winds) and the Aeolus L2C Rayleigh winds, i.e. LOS* winds based on ECMWF model data. Only Aeolus LOS* winds with
an estimated error below 4.8 m·s$^{-1}$ (HLOS: 8 m·s$^{-1}$) were considered valid. Due to the strong meridional wind especially in
the upper range gates of the A2D at the beginning of the common leg, large wind speed differences $\Delta > 5$ m·s$^{-1}$ were present
between Aeolus and the A2D which were compensated by the azimuth correction as explained above. Hence, the adapted
A2D Rayleigh winds show much better qualitative agreement with both the Aeolus Rayleigh winds and the model data.

## 4.3 Statistical comparison of A2D, Aeolus and ECMWF data

The adaptation of the A2D data to the Aeolus measurement grid and LOS pointing direction allowed for a statistical
comparison of the measured LOS* wind speeds. The scatterplots in Fig. 10 show the correlation of the three wind data sets
from the discussed underflight on 22 November 2018: the A2D Rayleigh winds, the Aeolus L2B Rayleigh winds and the
ECMWF model data (Fig. 10a). The scatter points are colour-coded with respect to the bottom altitude of the bin used for
comparison. The first plot shows good correspondence of the A2D winds with the model data for altitudes below 7 km, but
also reveals a positive mean bias of the A2D winds of 1.4 m·s$^{-1}$ which is evident over the entire wind speed range. Wind
speed differences above 2 m·s$^{-1}$ are especially present at higher altitudes (light-brown scatters). Since the accuracy of the
model data is assumed to be better than that of the A2D with a nearly vanishing bias and low random error around 2 m·s$^{-1}$
(Witschas et al., 2017; Witschas et al., 2019), it is used as the reference. The bias of the A2D winds is most likely related to
the incomplete telescope overlap close to the aircraft resulting in a reduced backscatter signal as well as a systematic wind
error (Paffrath et al., 2009). For the scatterplot shown in Fig. 10(a), the scaled MAD of 1.6 m·s$^{-1}$ is significantly larger than
the standard deviation (1.4 m·s$^{-1}$), indicating that the wind speed differences are not normally distributed, primarily owing to
the positively biased winds measured at higher altitudes. For the comparison of the Aeolus Rayleigh winds with the
ECMWF model data (Fig. 10(b)), the discrepancy is larger (standard deviation: 2.5 m·s$^{-1}$, scaled MAD: 2.0 m·s$^{-1}$) which is
mainly due to the two outliers that also become apparent in the Aeolus wind curtain (Fig. 9(a)) at about 1.5 km altitude.
Here, the small bin size of 250 m entails a poor signal-to-noise level, resulting in a large random wind error that is close to
the estimated error threshold of 4.8 m·s$^{-1}$ applied to the curtains in Fig. 9. The mean bias of the Aeolus winds compared to
the model is 0.5 m·s$^{-1}$. Consequently, when comparing the Aeolus to the A2D winds (see Fig. 10(c)), a negative bias
around -0.8 m·s$^{-1}$ is apparent. The small discrepancy of this value from the difference between the respective biases to the
ECMWF model (0.5 m·s$^{-1}$ – 1.4 m·s$^{-1}$ = -0.9 m·s$^{-1}$) can be explained with the different wind data coverage of the airborne
and satellite instrument which also results in a larger number of scatter points for the A2D-to-ECMWF comparison.
Regarding the statistical dispersion in the Aeolus-to-A2D scatterplot, the random errors of the two lidar instruments with
respect to the model winds approximately add up quadratically according to

$$\sigma_{\text{total}} \approx \sqrt{\sigma_{\text{A2D}}^2 + \sigma_{\text{Aeolus}}^2}. \tag{6}$$

This leads to a standard deviation of 3.0 m·s$^{-1}$ and a slightly smaller scaled MAD of 2.9 m·s$^{-1}$.

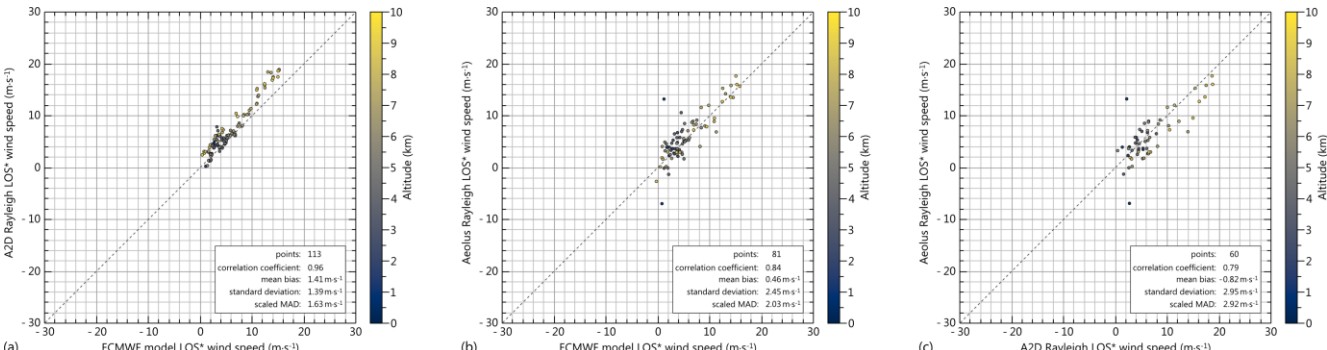

**Figure 10.** Scatterplots comparing (a) the A2D Rayleigh LOS* winds with the ECWMF model LOS* winds, (b) the Aeolus L2B Rayleigh LOS* winds with the ECWMF model LOS* winds and (c) the Aeolus L2B Rayleigh LOS* winds with the A2D Rayleigh LOS* winds for the wind scene on 22/11/2018 between 16:13 UTC and 17:15 UTC. The data points are colour-coded with respect to the bottom altitude of the respective bins used for comparison.

In analogy to the flight leg discussed above, the other collocated wind observations of the campaign listed in Table 2 were analysed. The resulting statistical comparison of the data sets from all underflights is shown in Fig. 11 and the statistical values derived from each scatterplot are summarised in Table 4. The latter also includes the parameters from the A2D-to-2-µm comparison discussed in section 3.4 after conversion to LOS* wind speeds. For the results from the comparison of the 2-µm DWL data with Aeolus and the ECWMF model winds please refer to Witschas et al. (2019).

When comparing the A2D to the model data from all flights (Fig. 11(a)), a mean bias of -0.9 m·s$^{-1}$ is calculated which is in fair agreement with the bias determined from the 2-µm reference lidar (-0.7 m·s$^{-1}$ LOS*) considering the smaller data overlap of the 2-µm DWL with the A2D compared to the model. The intercept of the linear regression function is even below -1 m·s$^{-1}$, while the slope deviates from the ideal case ($A = 1$) by 3%. This slope error is most likely related to an imperfect calibration of the Rayleigh channel. In particular, differences in atmospheric pressure and temperature encountered during the calibration procedure and the wind scene give rise to a mismatch between the derived calibration parameters and the actual Rayleigh channel behaviour during the underflight (Zhai et al., 2019).

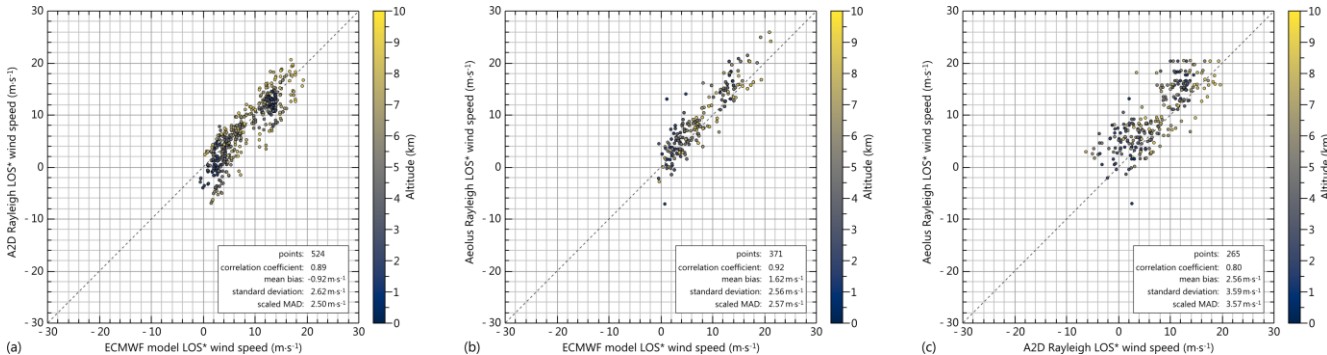

**Figure 11.** Scatterplots comparing (a) the A2D Rayleigh LOS* winds with the ECWMF model LOS* winds, (b) the Aeolus L2B Rayleigh LOS* winds with the ECWMF model LOS* winds and (c) the Aeolus L2B Rayleigh LOS* winds with the A2D Rayleigh LOS* winds for all underflights of the WindVal III campaign. The data points are colour-coded with respect to the bottom altitude of the respective bins used for comparison.





**Table 4.** Results of the statistical comparison between the A2D Rayleigh, the Aeolus Rayleigh and the ECMWF LOS* wind speeds for all underflights performed during the WindVal III campaign. See corresponding scatterplots in Fig. 11 (only last three columns). The first column includes the data from Table 3 after conversion of the wind speeds to the satellite's LOS (37° off-nadir angle).

| Statistical parameter | A2D Rayleigh vs. 2-µm DWL | A2D Rayleigh vs. ECMWF | Aeolus Rayleigh vs. ECMWF | Aeolus Rayleigh vs. A2D Rayleigh |
|---|---|---|---|---|
| Number of compared bins | 1301 | 524 | 371 | 265 |
| Correlation coefficient $r$ | 0.83 | 0.89 | 0.92 | 0.80 |
| Slope $A$ | $0.98 \pm 0.02$ | $1.03 \pm 0.03$ | $1.08 \pm 0.02$ | $0.83 \pm 0.04$ |
| Intercept $B$ | -0.7 m·s$^{-1}$ | -1.2 m·s$^{-1}$ | 0.9 m·s$^{-1}$ | 3.8 m·s$^{-1}$ |
| Mean bias | -0.6 m·s$^{-1}$ | -0.9 m·s$^{-1}$ | 1.6 m·s$^{-1}$ | 2.6 m·s$^{-1}$ |
| Standard deviation | 3.7 m·s$^{-1}$ | 2.6 m·s$^{-1}$ | 2.6 m·s$^{-1}$ | 3.6 m·s$^{-1}$ |
| Scaled MAD | 3.4 m·s$^{-1}$ | 2.5 m·s$^{-1}$ | 2.6 m·s$^{-1}$ | 3.6 m·s$^{-1}$ |

The scatterplot also shows that most of the over- and underestimated A2D winds are measured either at high altitudes (>8 km) or very close to the ground (<2 km). While the deviations close to the aircraft can be explained with the incomplete telescope overlap, the larger wind speed differences at lower altitudes are probably related to the influence of aerosols in the planetary boundary layer which cause Mie contamination of the Rayleigh signal and, in turn, introduce systematic errors of the winds measured in this region. Note that, in contrast to the A2D, a so-called cross-talk correction is performed for the satellite instrument to minimize such errors. Despite these error sources, the standard deviation of 2.6 m·s$^{-1}$ and scaled MAD of 2.5 m·s$^{-1}$ of the A2D Rayleigh winds are considerably lower than observed during previous airborne campaigns, as discussed above. The fact that the random error is even lower than the (LOS* converted) values obtained from the 2-µm DWL comparison (3.4 m·s$^{-1}$) can be explained by the coarser horizontal and vertical resolution of the model winds included in the L2C product which is provided on the same grid as the L2B product. Consequently, the number of A2D wind results which are averaged onto the model grid is larger than for the finer 2-µm grid (by a factor of $\approx 2.5$), thus reducing the variability in the bin-to-bin wind comparison (by a factor of $\approx \sqrt{2.5}$).

Comparing the Aeolus LOS* winds with the model data (Fig. 11(b)), a mean bias of 1.6 m·s$^{-1}$ and a random error of 2.6 m·s$^{-1}$ (scaled MAD) is derived. These values are in fair agreement with the results from the Aeolus-to-2-µm comparison described in Witschas et al. (2019) where a positive bias of 2.1 m·s$^{-1}$ (HLOS) and scaled MAD of 4.0 m·s$^{-1}$ (HLOS) was determined, corresponding to LOS* values of 1.3 m·s$^{-1}$ and 2.4 m·s$^{-1}$, respectively. A positive bias of the L2B Rayleigh winds (1.5 m·s$^{-1}$ HLOS) was also verified by comparative measurements using a ground-based wind lidar located in southern France in January 2019 (Khaykin et al., 2019). For the campaign time and region the bias of the L2B Rayleigh winds is beyond the mission requirements of Aeolus which should provide an accuracy of 0.7 m·s$^{-1}$ in HLOS winds (0.4 m·s$^{-1}$ LOS*) on a global scale, while the HLOS random error is required to be below 1 m·s$^{-1}$ (0.6 m·s$^{-1}$ LOS*) in the planetary boundary layer, 2.5 m·s$^{-1}$ (1.5 m·s$^{-1}$ LOS*) in the troposphere and 3 to 5 m·s$^{-1}$ (1.8 to 3.0 m·s$^{-1}$ LOS*) in the stratosphere in order to ensure a positive impact on the weather forecast by assimilating the wind data in numerical weather prediction models (ESA, 2016).





The wind bias is owed to the fact that the mission was still in the commissioning phase at the time of the campaign. In this period instrumental drifts were observed resulting in a long-term change of the incidence angle on the Rayleigh spectrometer and thus systematic Rayleigh wind errors. Also, the emit energy of the laser of 60 mJ was below the target value of 80 mJ which together with presumed losses in the receive path led to significantly lower signal-to-noise levels as expected (by about a factor of 2.5 to 3) (Reitebuch et al., 2019). As a consequence, the random error did not meet the system requirements

for the troposphere in the early phase of the mission. Considerable improvement of the accuracy and precision of the Aeolus data is foreseen by correction for instrumental drifts and by increasing the laser output energy via system adjustments, respectively. The systematic and random error of Aeolus can also be estimated from the Aeolus-to-A2D comparison depicted in Fig. 11(c). Here, the bias of 2.6 m·s$^{-1}$ translates to an actual bias of 2.6 m·s$^{-1}$ – 0.6 m·s$^{-1}$ = 2.0 m·s$^{-1}$ and 2.6 m·s$^{-1}$ – 0.9 m·s$^{-1}$ = 1.7 m·s$^{-1}$ when considering the negative bias of the A2D Rayleigh channel with respect to the 2-μm

DWL and the model, respectively. Using Eq. (6), the random error of the Aeolus winds are approximated to be $[(3.6 \text{ m·s}^{-1})^2 - [(2.6 \text{ m·s}^{-1})^2]^{1/2} = 2.5 \text{ m·s}^{-1}$. Hence, the A2D and Aeolus Rayleigh channels show very similar precision of about 2.5 m·s$^{-1}$, albeit the underlying reasons are of different nature.

**4.4 Influence of coverage ratio and mean distance thresholds**

The aerial weighted averaging algorithm described in section 4.1 holds the risk of large discrepancies between the averaged

A2D wind and the compared Aeolus L2B or ECMWF model wind from the L2C product in case that the measurement bin from the respective Aeolus data product is only sparsely covered by A2D bins, especially in regions with strong wind shear. An additional potential representativity error may arise from too large spatial separation between the A2D bins covering an Aeolus bin and the bin center of the latter. Therefore, two settable threshold parameters were defined for the statistical comparisons described in the previous section. While the minimum overlap of the compared bins (coverage ratio threshold,

CR) was chosen to be 25%, the upper threshold of the mean distance $d_{max}$ between the Aeolus bin center and the bin centers of all overlapping A2D bins was set to 40 km. The influence of the two parameters on the statistical parameters retrieved from the A2D-to-ECMWF comparison and the Aeolus-to-A2D comparison are shown in Fig. 12. The plots (a) and (c) illustrate that the choice of the coverage ratio has no significant effect on the bias and random error (<0.2 m·s$^{-1}$) for values below 50%. At higher thresholds the number of compared winds becomes considerably lower, resulting in stronger

dependence of the statistical values on the coverage ratio. Therefore, the results of the respective wind comparisons for which the number of compared bins is below 200 are considered statistically insignificant and indicated as grey-shaded areas in Fig. 12.

Regarding the maximum mean distance between the Aeolus L2B/L2C bin and the covering A2D bins (see Fig. 12(a) and (c)), a strong impact is observed for $d_{max}$ <30 km, as the number of compared winds is drastically decreased. Given the

horizontal length of the Aeolus bin of about 86 km, the statistical parameters remain constant for distances from the bin center above ≈40 km. From the above considerations, relaxed threshold parameters of CR = 25% and $d_{max}$ = 40 km were found to provide an optimal trade-off between comparability and an acceptable number of representative composite A2D





bins used for comparison. In this respect, the second threshold parameter $d_{max}$ was effectively switched-off to maximize the number of data points. However, adaptation of the two parameters is advisable for wind scenes exhibiting more

heterogeneous atmospheric conditions, as these are more prone to large representativity errors in case of sparse A2D data coverage. The same holds true for the comparison of Mie winds which is envisaged for future campaigns.

**Figure 12.** Influence of the (a,c) coverage ratio and (b,d) horizontal distance threshold on the statistical parameters resulting from the
A2D-to-ECMWF comparison (top) and the Aeolus-to-A2D comparison (bottom). The top panels of each subfigure depict the number of bins entering the statistical comparison; the middle panels show the bias and the bottom panels illustrate the random error (standard deviation and scaled MAD) depending on the respective threshold parameter. Results for which the number of compared bins is below 200 are considered statistically insignificant and thus indicated as grey-shaded areas.



### 4.5 Optimization of A2D range gate settings

It has to be stated that the statistical significance of the shown comparisons is rather limited, as there is only a small number of compared bins, especially for the Aeolus-to-A2D comparison (265). This is primarily due to the used vertical sampling of the A2D and Aeolus which had many small range gates in the lower troposphere (Fig. 13(a)), where Aeolus winds often exhibit large estimated wind errors above the threshold useable for comparison. The overlap of valid wind data from the two instruments was thus limited to the range between 2 km and 9 km where the vertical resolution of both wind lidars was set to

be coarser. With a view to the upcoming campaigns during the operational phase of the Aeolus mission, a more meaningful statistical comparison and hence better validation can be accomplished by adapting the vertical sampling strategy such that more small and medium-sized range gates are located at altitudes between 4 km and 8 km at the expense of lower resolution towards the ground. A proposed optimized A2D range gate setting is illustrated in Fig. 13(b) depicting exemplary Aeolus Rayleigh wind curtains from November 2018 and April 2019 with overlaid bin borders of the A2D (dashed lines) for an

aircraft flight altitude of 10675 m (35,000 feet, flight level 350). The diagram shows that the Aeolus range gate settings were already modified after the end of the commissioning phase in January 2019, providing higher resolution in the upper troposphere. By using range gates with 296 m and 592 m thickness for the A2D in the same region, the number of compared bins can be significantly increased. Furthermore, vertical sampling with higher resolution at altitudes close to the tropopause allows resolving jet streams that often reside in this region, thus delivering wind data over a wider wind speed range to enter

the statistical comparison. This will additionally improve the significance of the statistical comparison, as the error in the fit parameters derived from the linear regression will be reduced according to Eq. (2a). The tropopause region is of particular interest, as increased ECMWF model errors with respect to the 2-μm DWL were observed in recent studies (Schäfler et al., 2019).

In addition to the validation of the L2B wind speeds, the comparative analysis of the A2D and Aeolus wind results from

forthcoming airborne campaigns will rather be dedicated to case studies of special scenes. In this context, it is foreseen to investigate the different data coverage of the respective Rayleigh and Mie channels under various atmospheric conditions. Moreover, future studies will focus on the origin of large outliers in the L2B wind product that show a low estimated error but large deviations from the model. Here, the quality control mechanisms that have been developed and refined for the A2D over the past years can potentially be used to optimize the L2B processor algorithms.

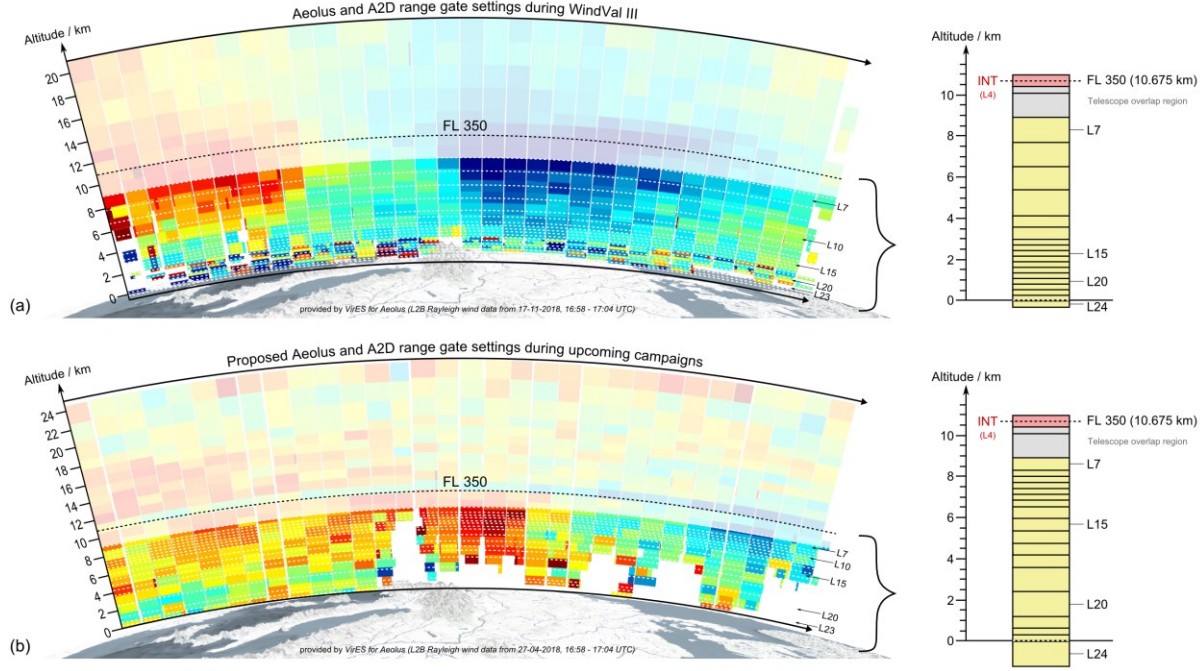

**Figure 13.** Diagram illustrating (a) the A2D range gate setting used during the WindVal III campaign and (b) an optimized range gate setting planned to be used in forthcoming airborne validation campaigns. The left part of the figure shows exemplary Aeolus L2B Rayleigh wind curtains with indicated bin altitudes of the A2D range gates (white dashed lines) assuming a flight altitude of about 10675 m (FL 350). The corresponding vertical range gates are additionally depicted on the right. Due to the incomplete telescope overlap, the wind data from the first 1.5 km below the aircraft shows increased wind errors and is thus not used. The figure was created based on screenshots from the Aeolus data visualization tool *VirES for Aeolus (https://aeolus.services/)*.

## 5 Summary and Conclusion

The airborne wind validation campaign WindVal III was carried out in Central Europe only three months after the launch of ESA's Earth Explorer mission Aeolus in August 2018. More than 3000 km of the Aeolus measurement swath were covered during four underflights with the DLR Falcon aircraft carrying the airborne demonstrator of the Aeolus payload as well as a coherent Doppler wind lidar. A2D data is available from three underflights adding up to more than 11 hours of wind measurements to be compared with the satellite data. Due to the sparse data coverage of the A2D and Aeolus Mie channels which was accepted in the flight planning for the benefit of better coverage of the respective Rayleigh channels, only the latter were further investigated in this study.

The WindVal III campaign has provided several lessons learned which will be considered in the forthcoming Cal/Val campaigns. Above all, it is envisaged to conduct dedicated flights with higher aerosol loading and larger cloud cover to allow for an assessment of the Mie channel performance. In particular, wind measurements in thin cirrus clouds are expected to yield valid Mie data over multiple range bins across the wind profile, as observed during flights in the North Atlantic region in the frame of the NAWDEX campaign in 2016 (Lux et al., 2016). Proper comparison of the A2D and Aeolus wind



results required adequate averaging and consideration of the different viewing geometries. An aerial interpolation algorithm was used for the adaptation of the A2D data to the Aeolus measurement grid, while conversion of the measured A2D LOS winds to the satellite LOS was realized with the aid of model wind data. The harmonized data sets were then compared to each other as well as to ECMWF model wind data which was used as a reference. Two settable threshold parameters were introduced to the algorithm and their influence on the correlation of the data sets was studied. The statistical comparison

revealed biases of -0.9 m·s$^{-1}$ and +1.6 m·s$^{-1}$ for the A2D and Aeolus LOS* Rayleigh wind speeds, respectively. The random errors were determined to be around 2.5 m·s$^{-1}$ on observation level for both direct-detection wind lidar instruments. The accuracy and precision of the A2D winds was significantly better compared to previous campaigns, whereas the Aeolus performance did not meet the formulated requirements of the mission for the studied wind scenes that took place during its commissioning phase. However, improvement of the satellite data quality is expected by a refinement the Aeolus processor

taking into account instrumental drift as well as by an enhancement of the laser output power. For future airborne validation campaigns an optimized range gate setting of the A2D will be implemented to increase the overlap with valid Aeolus wind data from the two instruments as well as to ensure a better sampling of strong wind gradients and higher wind speeds in the upper troposphere. Furthermore, a larger number of underflights will be performed for increasing the validity of the wind data comparison. This will also allow for various case studies which aim at the optimization of the Aeolus processor

algorithms.

*Data availability.* The A2D and 2-μm DWL data used in this paper can be provided upon request by email to Christian Lemmerz (Christian.Lemmerz@dlr.de). Aeolus data was obtained from the tool *VirES for Aeolus* (https://aeolus.services/).

*Competing interests.* The authors declare that they have no conflict of interest.

*Author contribution.* OL and CL conducted the A2D wind observations. CL was the PI of the WindVal III campaign. OL, FW and UM analysed the A2D data and developed the methodology to compare the data sets. BW conducted the 2-μm DWL measurements. SR processed the 2-μm DWL data. CL, OR and AG managed the WindVal III campaign and

conducted the flight planning. The paper was written by OL with contributions from all co-authors.

*Acknowledgements.* The development of the ALADIN Airborne Demonstrator and the work carried out during the WindVal III campaign were supported by the German Aerospace Center (Deutsches Zentrum für Luft- und Raumfahrt e.V., DLR) and the European Space Agency (ESA), providing funds related to the preparation of Aeolus (WindVal III, contract no.

4000114053/15/NL/FF/gp). The presented work includes preliminary data (not fully calibrated/validated and not yet publicly released) of the Aeolus mission that is part of the European Space Agency (ESA) Earth Explorer Programme. Further data quality improvements, including in particular a significant product bias reduction, will be achieved before the public data release. The analysis has been performed in the frame of the Aeolus Scientific Calibration & Validation Team (ACVT).



We thank Thorsten Fehr (Aeolus Campaign Coordinator) as well as Anne Grete Straume (Aeolus Mission Scientist) and
Jonas von Bismarck (Aeolus Data Quality Manager) for their support of the study. The first author was partly funded by a
young scientist grant by ESA within the DRAGON 4 program (contract no. 4000121191/17/I-NB). Finally, the authors are
especially grateful to Engelbert Nagel for his constant technical assistance throughout the campaign. The support of the DLR
flight facility for the realization of the airborne campaign is greatly acknowledged as well.

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
