# Peer review of "Intercomparison of wind observations from ESA's satellite mission Aeolus and the ALADIN Airborne Demonstrator"

_Atmospheric Measurement Techniques, 2019_

## Referee Comment (RC1) · Anonymous Referee #2 · 30 Jan 2020

**Review report for manuscript amt-2019-431**

The current study deals with the intercomparison of AEOLUS' wind observations versus the ALADIN airborne demonstrator, whereas meteorological numerical outputs from the ECMWF are also employed for the further assessment of the spaceborne and airborne wind profiles. The analysis has been performed in the framework of the WindVal III campaign in which flights of the DLR Falcon are collocated with AEOLUS L2B observations. The manuscript is well organized and written, presenting adequately the obtained results while the authors' recommendations for relevant future Cal/Val studies enhance the quality of their work. The topic of the submitted paper fits very well to the scientific purposes of the AMT and can be published after addressing some minor comments and suggestions which are listed below.

1. I think that it will be useful to provide a figure with the AEOLUS' observational geometry in order to help the readers to understand better the LOS, HLOS, projections etc.
2. My opinion is that much of the technical details (Sections 2.1 and 3.1) can be removed from the text.
3. How much independent can be the comparison between AEOLUS and ECMWF winds since the spaceborne data are used in the data assimilation of the model?
4. Line 168: Clarify that groups refer to the observation level (i.e., Lines 146-147).
5. Line 170: Could you be more specific about the estimates of the scattering ratio?
6. Apart from demonstrator, ALADIN and model winds, have you checked if there are available data from soundings (e.g. Wyoming)?
7. Line 180: Can you provide a short statement about the "*known error sources*"?
8. Line 183: I would just say u and v components of the wind vector.
9. Line 187: Replace "*…, Germany in the in the time frame…*" with "*…, Germany in the time frame…*".
10. Lines 239-241: Mie signals attributed to aerosols' presence can also "contaminate" the Rayleigh response.
11. In all curtain plots the longitudes are missing in x axes.
12. Line 321: Do you mean below the clouds?
13. Equation 1: If I am not missing something, the formula should be LOS=HLOS/sin(Θ). Please see my first comment.
14. Lines 431-432: Rephrase this sentence because it is somehow misleading. I suppose that the aerial weighting it is applied both in vertical and horizontal terms.
15. Lines 500-503: It would be useful to provide also the curtain plot for the A2D winds without the off-nadir angle correction.
16. Lines 514-515: It is quite strange to consider model outputs as reference values!
17. Figures 10 and 11: How much different are the results when are considered only coincident measurements from the three datasets?

---

## Referee Comment (RC2) · Anonymous Referee #1 · 8 Feb 2020

The paper by Oliver Lux and coauthors reports the results of an airborne campaign dedicated to Aeolus wind product validation using its airborne demonstrator A2D deployed onboard Falcon aircraft. The presented cal/val experiment is an important contribution and a tremendous effort towards improvement of direct-detection sensing of wind from space. The paper is carefully written, the experimental setup and validation methodology are thoroughly and comprehensively described, the graphical material is prepared with care, whereas the conclusions and recommendations are well substantiated. That said, the general issue of this paper, in my opinion, is that its content is excessively biased towards methodological aspects of the intercomparison. I assume that most potential readers of this article would be rather interested in the Aeolus

validation part (since the airborne lidars and their performance are already described elsewhere), however they would have to read a long way to get to what they are looking for. The title of the paper promises a long-awaited result of Aeolus validation using its airborne demonstrator within a dedicated validation campaign carried out by the renown experts in the off-ground lidar technique. However, the results of A2D-Aeolus validation are mixed together with the A2D-ECMWF-AEOLUS statistical figures. It gets worse when the key results of intercomparison are reported as lidars' biases with respect to ECMWF. This raises a valid question i.e. what the Falcon-Aeolus underflights are for, if both lidars are finally referenced to the model. I recommend the authors to address the following remarks, in order to reconcile the inconsistencies between the title and the content.  c In the introduction (l.57-58), the authors claim their study a methodological reference for the airborne experiments on Aeolus validation. With that, the cal/val experiment is restricted to Rayleigh wind measurements. What are the other clear-air airborne Doppler lidars involved in Aeolus validation? If there are none, this methodological reference could be restricted to internal use. Please be more specific regarding the scope of potential applications of the presented methodology.  c The key example of Aeolus-A2D intercomparison is presented in Fig. 9, however the panels a) and c) are difficult to compare as they are interspersed by the panel 9b, which should belong to the section describing the intercomparison setup.  c Apart from the spatial curtains in Fig. 9, it would be useful to show a few examples of individual wind profiles measured by both lidars, probably also at their native vertical resolution This will give a much better feeling on the capacities of different lidars than the tabulated numbers.  c The results of A2D-Aeolus intercomparison should be presented in a separate section devoted to lidar-lidar intercomparison. The key figures of Aeolus-A2D intercomparison statistics (which is the title of the paper) should be provided in the abstract and conclusions. The comparison against ECMWF should be reported in a specific subsection of the manuscript.  c The discussion on the representability of the Aeolus cal/val results could be better developed in the context of the preliminary nature of L2B wind product.

---

## Referee Comment (RC3) · Anonymous Referee #1 · 8 Feb 2020

**Review of Lux et al. 2019 "Intercomparison of wind observations from ESA's satellite mission Aeolus and the ALADIN Airborne Demonstrator"**

The paper by Oliver Lux and coauthors reports the results of an airborne campaign dedicated to Aeolus wind product validation using its airborne demonstrator A2D operating onboard Falcon aircraft together with DWL lidar. The presented cal/val experiment is an important contribution and a tremendous effort towards improvement of direct-detection sensing of wind from space. The paper is carefully written, the experimental setup and validation methodology are thoroughly and comprehensively described, the graphical material is prepared with care, whereas the conclusions and recommendations are well substantiated.

That said, the general issue of this paper, in my opinion, is that its content is excessively biased towards methodological aspects of the intercomparison. I assume that most potential readers of this article would be rather interested in the Aeolus validation part (since the airborne lidars and their performance are already described elsewhere), however they would have to read a long way to get to what they are looking for.

The title of the paper promises a long-awaited result of Aeolus validation using its airborne demonstrator within a dedicated validation campaign carried out by the renown experts in the off-ground lidar technique. However, the results of A2D-Aeolus validation are mixed together with the A2D-ECMWF-AEOLUS statistical figures. It gets worse when the key results of intercomparison are reported as lidars' biases with respect to ECMWF. This raises a valid question i.e. what the Falcon-Aeolus underflights are for, if both lidars are finally referenced to the model.

I recommend the authors to address the following remarks, in order to reconcile the inconsistencies between the title and the content.

- In the introduction (l.57-58), the authors claim their study a methodological reference for the airborne experiments on Aeolus validation. With that, the cal/val experiment is restricted to Rayleigh wind measurements. What are the other clear-air airborne Doppler lidars involved in Aeolus validation? If there are none, this methodological reference could be restricted to internal use. Please be more specific regarding the scope of potential applications of the presented methodology.
- The key example of Aeolus-A2D intercomparison is presented in Fig. 9, however the panels a) and c) are difficult to compare as they are interspersed by the panel 9b, which should belong to the section describing the intercomparison setup.
- Apart from the spatial curtains in Fig. 9, it would be useful to show a few examples of individual wind profiles measured by both lidars, probably also at their native vertical resolution This will give a much better feeling on the capacities of different lidars than the tabulated numbers.
- The results of A2D-Aeolus intercomparison should be presented in a separate section devoted to lidar-lidar intercomparison. The key figures of Aeolus-A2D intercomparison statistics (which is the title of the paper) should be provided in the abstract and conclusions. The comparison against ECMWF should be reported in a specific subsection of the manuscript.

- The discussion on the representability of the Aeolus cal/val results could be better developed in the context of the preliminary nature of L2B wind product.

---

## Author Comment (AC1) · 20 Feb 2020

***Response to Referee Comment (RC1) on***

*Intercomparison of wind observations from ESA's satellite mission Aeolus*
*and the ALADIN Airborne Demonstrator ([https://doi.org/10.5194/amt-2019-431](https://doi.org/10.5194/amt-2019-431))*

We are grateful for the referee's very valuable and positive comments on our manuscript. Following the suggestions and questions, the following aspects will be elaborated more in detail in a revised version of the paper.

General comment:

*The current study deals with the intercomparison of AEOLUS' wind observations versus the ALADIN airborne demonstrator, whereas meteorological numerical outputs from the ECMWF are also employed for the further assessment of the spaceborne and airborne wind profiles. The analysis has been performed in the framework of the WindVal III campaign in which flights of the DLR Falcon are collocated with AEOLUS L2B observations. The manuscript is well organized and written, presenting adequately the obtained results while the authors' recommendations for relevant future Cal/Val studies enhance the quality of their work. The topic of the submitted paper fits very well to the scientific purposes of the AMT and can be published after addressing some minor comments and suggestions which are listed below.*

Comment #1.1:

*I think that it will be useful to provide a figure with the AEOLUS' observational geometry in order to help the readers to understand better the LOS, HLOS, projections etc.*

Response to Comment #1.1:

A new subfigure (Fig. 5(a)) was added to the manuscript, depicting the Aeolus observational geometry and the wind vector projections (see below). The figure is referenced in the context of Eq. (1), where the relationship between LOS and HLOS winds is introduced (see also response to Comment 1.13).

[Figure]

[Figure]

**Figure 1.** (a) Aeolus observational geometry (b) Aeolus L2B LOS* Rayleigh winds (positive if winds are blowing away from the instrument) measured during the underflight on 22 November 2018 between 40.6°N and 47.2°N. Only winds with an estimated wind error of less than 12 m·s⁻¹ are shown. Winds at altitudes above 10 km are outside of the measurement range of the A2D and therefore shown greyed out. The figure was created based on a screenshot from the Aeolus visualization tool *VirES for Aeolus* *(https://aeolus.services/)*.

Comment #1.2:

*My opinion is that much of the technical details (Sections 2.1 and 3.1) can be removed from the text.*

Response to Comment #1.2:

We agree that the first part of the paper, especially the instrument description of the A2D (section 2.1) and the explanation of the response calibrations (section 3.1), are too long given the main focus of the article, namely the intercomparison of the A2D wind results with those of Aeolus. Therefore, we have shortened the two sections mentioned above to more concentrate on the methodology for adapting the A2D wind data to the Aeolus grid and viewing angle and on the wind comparisons. Nevertheless, we believe that it is important for the readers to understand the differences between the airborne and the satellite-borne wind lidar regarding the design and data acquisition, since these aspects are crucial for the understanding of the respective error sources and related limitations in terms of accuracy and precision.

In the revised manuscript, technical details in sections 2.1 and 3.1 were removed from the text. In particular, detailed information on the specifications of the receiver spectrometers and the measurement principle were omitted. Moreover, the explanation of the response calibrations was shortened.

Comment #1.3:

*How much independent can be the comparison between AEOLUS and ECMWF winds since the spaceborne data are used in the data assimilation of the model?*

Response to Comment #1.3:

As of November 2018, the Aeolus data was not yet used in the data assimilation of the model. Therefore, the two datasets are uncorrelated and the statistical comparison between them is independent. Operational assimilation of the Aeolus data by the ECMWF has started on 9 January 2020.

Comment #1.4:

*Line 168: Clarify that groups refer to the observation level (i.e., Lines 146-147).*

Response to Comment #1.4:

The grouping of the L2B processor was specified in revised manuscript as follows:

As a first step, Aeolus measurements (horizontal resolution of about 2.9 km corresponding to 0.4 s) are gathered together into groups where the length depends on the L2B parameter settings. During the analysed period in November 2018 the group length was set to 30 Aeolus measurements, **and thus identical to the previously defined observation length**, corresponding to a horizontal extent of about 86.4 km. **Note that groups can also be shorter than observations in case the horizontal averaging is set differently in the L2B processor.**

Comment #1.5:

*Line 170: Could you be more specific about the estimates of the scattering ratio?*

Response to Comment #1.5:

The distinction between "clear" and "cloudy" was specified in the revised manuscript as follows:

The measurement bins within the group are then classified into "clear" and "cloudy" bins using estimates of the **backscatter ratio which is defined as the ratio of the total backscatter coefficient (particles and molecules) to the molecular backscatter coefficient. "Clear" bins are usually those for which the backscatter ratio is below 1.2 to 1.4 depending on L2B processor settings, while bins with higher backscatter ratios are considered "cloudy".**

Comment #1.6:

*Apart from demonstrator, ALADIN and model winds, have you checked if there are available data from soundings (e.g. Wyoming)?*

Response to Comment #1.6:

During the WindVal III campaign, only the first satellite underflight on 17 November 2018 included a ground station overpass from which wind data could be obtained for additional comparison. In particular, the ground station in Nordholz, Germany (53.78°N, 8.67° E) which operates a radar wind profiler was passed twice at 17:34 UTC and 17:59 UTC. However, since the A2D was not operational during this underflight, comparisons of the wind profiler data with the other lidar instruments were not performed. Regarding the soundings available from the website of the University of Wyoming, the spatial and temporal separation from the Aeolus wind observations is too large (>100 km and /or >1 hour) to allow for a meaningful comparison of the wind data. The analysis of collocated wind observations of Aeolus with radar wind profilers and radiosondes is carried out by other Cal/Val teams and not reported here. In addition, validation of the Aeolus winds by means of the coherent 2-µm Doppler wind lidar is comprehensively discussed in Witschas et al. (2020).

Comment #1.7:

*Line 180: Can you provide a short statement about the "known error sources"?*

Response to Comment #1.7:

The "known error sources" are briefly specified in the revised manuscript as follows:

Aeolus wind data obtained from the L2B product which is discussed here is in a preliminary state, inasmuch as biases related to known error sources **such as instrumental drifts** are not corrected yet (Reitebuch et al., 2019; Rennie and Isaksen, 2019a).

A more comprehensive discussion of the error sources is included in section 4.3.

Comment #1.8:

*Line 183: I would just say u and v components of the wind vector.*

Response to Comment #1.8:

The text was changed accordingly to:

It contains the ***u* and *v* components of the wind vector** and supplementary geophysical parameters.

Comment #1.9:

*Line 187: Replace "…, Germany in the in the time frame…" with "…, Germany in the time frame…"*

Response to Comment #1.9:

The sentence was corrected. Thanks for noticing.

Comment #1.10:

*Lines 239-241: Mie signals attributed to aerosols' presence can also "contaminate" the Rayleigh response.*

Response to Comment #1.10:

The sentence was changed to

Above all, **cloud- and aerosol-free** conditions are necessary to avoid Mie backscatter signals which affect the backscatter spectrum, and thus **contaminate** the Rayleigh response in the respective range gates.

Comment #1.11:

*In all curtain plots the longitudes are missing in x axes.*

Response to Comment #1.11:

Longitudes were added to the x-axes of all curtain plots in Figs. 3, 4 and 9. Please note that, in contrast to the latitudes, the longitudes did not linearly increase with the flight time during the underflight on 22 November (see also flight track on Fig. 8), so that we refrained from showing an additional longitude axis with non-equidistant tick marks. Instead, for each integer latitude on the x-axes the corresponding longitude is provided in parentheses.

Comment #1.12:

*Line 321: Do you mean below the clouds?*

Response to Comment #1.12:

That's correct. The text was changed to:

Wind data is mainly obtained from the cloud tops along the track. Due to the high optical density of the clouds, the laser was strongly attenuated, thus preventing sufficient backscatter signal and valid Mie wind data over multiple range gates **within and below** the clouds.

Comment #1.13:

*Equation 1: If I am not missing something, the formula should be LOS=HLOS/sin(Θ). Please see my first comment.*

Response to Comment #1.13:

Equation (1) is correct, as can be derived from the shown wind vector projections shown in the added subfigure Fig. 5(a), see response to Comment 1.1.

Comment #1.14:

*Lines 431-432: Rephrase this sentence because it is somehow misleading. I suppose that the aerial weighting it is applied both in vertical and horizontal terms.*

Response to Comment #1.14:

The sentence was modified as follows:

Each valid A2D range bin covering an Aeolus range bin is allocated **both** horizontal and vertical weights depending on the size of **its** contribution to the total area of the Aeolus bin, as illustrated in Fig. 7.

Comment #1.15:

*Lines 500-503: It would be useful to provide also the curtain plot for the A2D winds without the off-nadir angle correction.*

Response to Comment #1.15:

A curtain plot without off-nadir angle correction, albeit at the original horizontal and vertical resolution of the A2D, is already provided in Fig. 4(a). We believe that there is no large benefit in presenting the same plot after adaptation to the Aeolus grid, but without applying the off-nadir angle correction, especially as the latter is effectively a simple scaling of the wind speeds by the factor $\sin(37°)/\sin(20°) \approx 1.76$.

Comment #1.16:

*Lines 514-515: It is quite strange to consider model outputs as reference values!*

Response to Comment #1.16:

We agree that, in principle, model output is usually not considered as a reference for observational data. However, comparison of the ECMWF model winds (averaged onto the Aeolus measurement grid) with data from the highly-accurate 2-μm coherent Doppler wind lidar showed a nearly vanishing bias and low random error around 2 m·s$^{-1}$, while the systematic and random errors of the A2D (-0.7 m·s$^{-1}$, 3.4 m·s$^{-1}$) and Aeolus Rayleigh winds (1.3 m·s$^{-1}$, 2.4 m·s$^{-1}$) with respect to the 2-μm DWL, determined on the A2D and Aeolus measurement grids respectively, were considerably

larger for the reasons described in the manuscript. Therefore, the model data was considered as the reference and also plotted on the x-axes of the scatterplots in Figs. 10 and 11.

Comment #1.17:

*Figures 10 and 11: How much different are the results when are considered only coincident measurements from the three datasets?*

Response to Comment #1.17:

When considering only those bins in the statistical comparisons for which valid wind data is available in all three datasets (ECMWF, A2D, Aeolus), i.e. considering only their common overlap, the statistical results from the model comparisons are slightly different. The corresponding scatterplots (in analogy to Figs. 10 and 11) are shown below.

[Figure]

Scatterplots comparing (a) the A2D Rayleigh LOS* winds with the ECWMF model LOS* winds, (b) the Aeolus L2B Rayleigh LOS* winds with the ECWMF model LOS* winds and (c) the Aeolus L2B Rayleigh LOS* winds with the A2D Rayleigh LOS* winds for the wind scene on 22/11/2018 between 16:13 UTC and 17:15 UTC. The data points are colour-coded with respect to the bottom altitude of the respective bins used for comparison. Only bins for which valid wind data is available in all three datasets (ECMWF, A2D, Aeolus) are considered.

[Figure]

Scatterplots comparing (a) the A2D Rayleigh LOS* winds with the ECWMF model LOS* winds, (b) the Aeolus L2B Rayleigh LOS* winds with the ECWMF model LOS* winds and (c) the Aeolus L2B Rayleigh LOS* winds with the A2D Rayleigh LOS* winds for all underflights of the WindVal III campaign. The data points are colour-coded with respect to the bottom altitude of the respective bins used for comparison. Only bins for which valid wind data is available in all three datasets (ECMWF, A2D, Aeolus) are considered.

For the wind scene on 22 November 2018, the A2D Rayleigh winds exhibit a slightly lower bias of 1.35 m·s$^{-1}$ (instead of 1.41 m·s$^{-1}$) with respect to the ECMWF model winds, whereas the bias of the Aeolus Rayleigh winds is slightly larger (0.53 m·s$^{-1}$ instead of 0.46 m·s$^{-1}$). The respective random errors (in terms of the standard deviation) are comparable as well (1.4 m·s$^{-1}$ for A2D, 2.6 m·s$^{-1}$ for Aeolus). Consequently, when cross-comparing the two lidar datasets on their coverage overlap, the bias of the Aeolus winds with respect to the A2D winds is 0.53 m·s$^{-1}$ – 1.35 m·s$^{-1}$ = -0.82 m·s$^{-1}$, while the random errors add up quadratically: $[(1.4\ \text{m·s}^{-1})^2 + (2.6\ \text{m·s}^{-1})^2] \approx 3.0\ \text{m·s}^{-1}$.

Regarding the datasets from the entire WindVal III campaign, the A2D Rayleigh wind bias is nearly unaffected (-0.87 m·s$^{-1}$ instead of -0.92 m·s$^{-1}$), when restricting the model comparison to those bins where valid Aeolus Rayleigh bins are also available. The same hold true for the Aeolus wind bias with respect to the ECMWF model data (1.68 m·s$^{-1}$ instead of 1.62 m·s$^{-1}$). As a result, the Aeolus-to-A2D comparison yields a bias of 1.68 m·s$^{-1}$ – (-0.87 m·s$^{-1}$) $\approx$ 2.56 m·s$^{-1}$. The respective random errors are only changed by around 0.2 m·s$^{-1}$.

Taking into account the limited wind data obtained from the three underflights of the WindVal III campaign which is also a result from the A2D and Aeolus range gate settings, as explained in section 4.5 of the manuscript, we think that it is beneficial not to restrict the model comparisons of the two lidar instruments to the common overlap of all three datasets. In this way, the already small number of compared winds is not further reduced, although the cross-comparison of the three datasets is more complicated.

The following sentence was added to the discussion section of the wind comparisons:

**It should be noted that the statistical results from the mutual comparisons only slightly deviate from the shown values (by less than 0.2 m·s$^{-1}$) when restricting the respective datasets to those bins where both instruments have valid wind data.**

---

## Author Comment (AC2) · 20 Feb 2020

***Response to Referee Comment (RC2) on***

*Intercomparison of wind observations from ESA's satellite mission Aeolus*
*and the ALADIN Airborne Demonstrator (https://doi.org/10.5194/amt-2019-431)*

We appreciate the referee's very insightful and helpful remarks on our manuscript. The responses to the individual comments and the corresponding changes in the manuscript are presented in the following.

General Comment:

*The paper by Oliver Lux and coauthors reports the results of an airborne campaign dedicated to Aeolus wind product validation using its airborne demonstrator A2D operating onboard Falcon aircraft together with DWL lidar. The presented cal/val experiment is an important contribution and a tremendous effort towards improvement of direct-detection sensing of wind from space. The paper is carefully written, the experimental setup and validation methodology are thoroughly and comprehensively described, the graphical material is prepared with care, whereas the conclusions and recommendations are well substantiated.*

*That said, the general issue of this paper, in my opinion, is that its content is excessively biased towards methodological aspects of the intercomparison. I assume that most potential readers of this article would be rather interested in the Aeolus validation part (since the airborne lidars and their performance are already described elsewhere), however they would have to read a long way to get to what they are looking for.*

*The title of the paper promises a long-awaited result of Aeolus validation using its airborne demonstrator within a dedicated validation campaign carried out by the renown experts in the off-ground lidar technique. However, the results of A2D-Aeolus validation are mixed together with the A2D-ECMWF-AEOLUS statistical figures. It gets worse when the key results of intercomparison are reported as lidars' biases with respect to ECMWF. This raises a valid question i.e. what the Falcon-Aeolus underflights are for, if both lidars are finally referenced to the model.*

*I recommend the authors to address the following remarks, in order to reconcile the inconsistencies between the title and the content.*

Response to the General Comment:

We agree that there are inconsistencies between the title and the content of the paper in its originally submitted version, which primarily originate from the mixture of the results from the Aeolus validation using the A2D with model comparisons of both lidar instruments. We also share the opinion that the first part of the paper, especially the instrument description of the A2D (section 2.1) and the explanation of the response calibrations (section 3.1), are too long given the main focus of the article, namely the intercomparison of the A2D wind results with those of Aeolus. Therefore, we have shortened the two sections mentioned above to more concentrate on the methodology for adapting the A2D wind data to the Aeolus grid and viewing angle and on the wind comparisons. Nevertheless, we believe that it is important for the readers to understand the differences between the airborne and the satellite-borne wind lidar regarding the design and data acquisition, since these aspects are crucial for the understanding of the respective error sources and related limitations in terms of accuracy and precision. Furthermore, the assessment of the systematic and random error of the A2D by comparison with the 2-μm DWL and the ECMWF model is a prerequisite for the later derivation of the Aeolus accuracy and precision. This approach is not clearly formulated in the original version of the manuscript, but better presented in the revised version. In particular, following the referee's suggestion, we have separated the model comparisons from the lidar-lidar comparison (see also response to comment #2.4 below). Overall, we regard this paper as the basis for upcoming publications that will focus on the Aeolus validation during the operational phase of the mission. It is thus intended to present the necessary tools to be applied for making the wind data from the A2D (and other wind lidars) comparable with those of Aeolus, rather than to provide an extensive validation study. The latter will be the subject of forthcoming publications dealing with the airborne campaigns conducted in May and September 2019 which also yielded a much larger data set.

In order to remedy the deficiencies of the paper, we have revised it according to the reviewer's comments. The responses to the individual comments and the related changes to the manuscript are elaborated below.

Comment #2.1:

*In the introduction (l.57-58), the authors claim their study a methodological reference for the airborne experiments on Aeolus validation. With that, the cal/val experiment is restricted to Rayleigh wind measurements. What are the other clear-air airborne Doppler lidars involved in Aeolus validation? If there are none, this methodological reference could be restricted to internal use. Please be more specific regarding the scope of potential applications of the presented methodology.*

Response to Comment #2.1:

The methodology described in the manuscript is not only applicable (and necessary) for the A2D, but also any other Cal/Val instrument that measures only one LOS component of the wind vector. One example is the *LEANDRE New Generation* (LNG) which was developed at the *Laboratoire Atmosphères, Milieux, Observations Spatiales* (LATMOS). The three-wavelength-dual-polarization-backscatter lidar is based on a two-wave Mach–Zehnder interferometer and is, amongst others, capable of measuring line-of-sight wind speeds (Bruneau et al, 2015). The instrument is foreseen to be deployed on airborne campaigns for the Aeolus validation and was already used in pre-launch campaigns during NAWDEX (Schäfler et al., 2018) on coordinated flights with the DLR Falcon. Intercomparison of the LNG wind data with the Aeolus winds will necessitate the consideration of the different viewing geometries, as it is the case for the A2D. In this respect, the study can be regarded as a methodological reference for upcoming publications on the Aeolus validation by means of the A2D, LNG or other wind lidars without the capability to retrieve the entire wind vector.

The introduction was specified along these lines in order to stress the relevance of the shown methodology:

**More specifically, it is shown how to take account of the different LOS directions in order to make the wind data sets comparable. This procedure is not only required for the A2D, but any other Cal/Val instrument that measures only one component of the wind vector, such as e.g. the LEANDRE New Generation (LNG) (Bruneau et al., 2015) which is also foreseen to be deployed on airborne campaigns for the Aeolus validation.**

The following reference was added:

**Bruneau, D., Pelon, J., Blouzon, F., Spatazza, J., Genau, P., Buchholtz, G., Amarouche, N., Abchiche, A., and Aouji, O.: 355-nm high spectral resolution airborne lidar LNG: System description and first results, Appl Opt, 54, 8776–8785, doi:10.1364/AO.54.008776, 2015.**

Moreover, the following sentence was added to the conclusions section:

**This procedure is not only of relevance for future validation campaigns employing the A2D, but also other wind lidars without the capability to retrieve the entire wind vector.**

Comment #2.2:

*The key example of Aeolus-A2D intercomparison is presented in Fig. 9, however the panels a) and c) are difficult to compare as they are interspersed by the panel 9b, which should belong to the section describing the intercomparison setup.*

Response to Comment #2.2:

See Response to Comment #2.3.

Comment #2.3:

*Apart from the spatial curtains in Fig. 9, it would be useful to show a few examples of individual wind profiles measured by both lidars, probably also at their native vertical resolution. This will give a much better feeling on the capacities of different lidars than the tabulated numbers.*

Response to Comment #2.3:

We agree with the reviewer's comments #2.2 and #2.3, and have revised Fig. 9 of the manuscript (see below). The panels (a) to (d) have been re-sorted such that the original A2D Rayleigh wind curtain after adaptation to the Aeolus grid but without azimuth correction is placed on the left-hand side of the figure together with the ECMWF model winds (from the Aeolus L2C product). The latter serve to apply the azimuth correction, as explained in section 4.1 of the manuscript, to produce the azimuth-corrected A2D wind curtain shown in panel (c). The Aeolus L2B Rayleigh wind curtain is plotted just below (panel (d)) which facilitates the comparison of the two. Moreover, the wind profile for one selected Aeolus observation is shown in an added panel (e) together with the corresponding profiles of the other three datasets. Here, the error bar for the Aeolus winds (blue squares) represents the estimated error included in the L2B product, while the error bar for the azimuth-corrected A2D winds (green dots) corresponds to the weighted standard deviation of the A2D winds from those bins that overlap with the respective Aeolus bin. Comparison of the wind profiles not only demonstrates the necessity of the azimuth correction, but also shows the good agreement of the Aeolus winds with the A2D and model data within the error margins.

[Figure]

**Figure 1.** LOS* wind profiles obtained during the underflight on 22 November 2018 between 40.5°N and 47.2°N: (a) A2D Rayleigh winds averaged onto the Aeolus measurement grid and for an off-nadir angle of 37°, but without azimuth correction, (b) ECMWF model winds, (c) A2D Rayleigh winds with azimuth correction and (d) Aeolus L2B Rayleigh winds. White colour represents missing or invalid data of one of the two instruments, e.g. below dense clouds. Only Aeolus Rayleigh LOS* winds with an estimated error below 4.8 m·s⁻¹ were considered valid. The wind profile for one selected Aeolus observation is shown in panel (e) together with the corresponding profiles of the other three datasets. The error bar for the Aeolus winds (blue squares) represents the estimated error included in the L2B product, while the error bar for the azimuth-corrected A2D winds (green dots) corresponds to the weighted standard deviation of all A2D bins contributing to the respective Aeolus bin. For the uncorrected A2D winds (grey dots), the error bars were omitted for the sake of clarity.

We refrained from plotting the A2D wind profile in its original vertical resolution, since the azimuth correction procedure, in its current state, is only applicable after adaptation of the A2D data onto Aeolus grid. This is because the Aeolus L2C data (ECMWF model output on Aeolus measurement track), in particular the $u$ and $v$ components, are provided on that same grid. Hence, either higher resolution model data or interpolation of the L2C data onto the A2D grid would be required to apply the azimuth correction to the A2D data in its native resolution. Furthermore, since there are on average around 20 A2D observations within the range of one Aeolus observation, we think that it is more representative to show the weighted wind speed average of all observations instead of selecting one particular observation out of the set of 20.

The modified version of Fig. 9 was included in revised manuscript and discussed in section 4.2 as follows:

**The left part of the figure shows the A2D Rayleigh winds averaged onto the Aeolus measurement grid and for an off-nadir angle of 37°, but without azimuth correction (panel (a)) and the Aeolus L2C Rayleigh winds, i.e. LOS* winds based on ECMWF model data (from the Aeolus L2C product) (b). The A2D Rayleigh winds after azimuth correction (A2D LOS* winds) (c) and the Aeolus L2B Rayleigh winds (d) are depicted in the middle of Fig. 9.**

**Moreover, the wind profile for one selected Aeolus observation is shown in panel (e) together with the corresponding profiles of the other three datasets. Here, the error bar for the Aeolus winds (blue squares) represents the estimated error included in the L2B product, while the error bar for the azimuth-corrected A2D winds (green dots) corresponds to the weighted standard deviation of the A2D winds from those bins that overlap with the respective Aeolus bin. Only Aeolus LOS* winds with an estimated error below 4.8 m·s$^{-1}$ (HLOS: 8 m·s$^{-1}$) were considered valid. Comparison of the curtain plots and the selected wind profiles demonstrates the necessity of the azimuth correction. Due to the strong meridional wind especially in the upper range gates of the A2D at the beginning of the common leg, large wind speed differences $\Delta > 5$ m·s$^{-1}$ were present between Aeolus and the uncorrected A2D data (grey dots) which were compensated by the azimuth correction as explained above. Hence, the adapted A2D Rayleigh winds show much better agreement with both the Aeolus Rayleigh winds and the model data. The weighted standard deviation of the A2D winds, indicated by the error bars, represents a measure of the variability of the A2D winds within the compared Aeolus bin. The values are on the order of 2 to 4 m·s$^{-1}$ and determined by both the random error of the A2D as well as the horizontal and vertical wind gradients within the respective Aeolus bin.**

Comment #2.4:

*The results of A2D-Aeolus intercomparison should be presented in a separate section devoted to lidar-lidar intercomparison. The key figures of Aeolus-A2D intercomparison statistics (which is the title of the paper) should be provided in the abstract and conclusions. The comparison against ECMWF should be reported in a specific subsection of the manuscript.*

Response to Comment #2.4:

Following the reviewer's suggestion, we have separated the model comparisons (section 4.3) from the lidar-lidar intercomparison (section 4.4). For this purpose, the figures showing the scatterplots from the three-dataset comparison for the selected wind scene on 22 November 2018 (Fig. 10) and the entire campaign (Fig. 11) were re-sorted. In the revised manuscript, there are three figures with two scatterplots each, depicting the A2D-to-model, the Aeolus-to-model and the Aeolus-to-A2D statistical results for the selected wind scene and the entire campaign, respectively. The discussion of the mutual comparisons has been adapted to the new structure accordingly.

In addition, the key figures of the Aeolus-to-A2D comparison statistics were added to the abstract and conclusions. In particular, the abstract of the revised manuscript includes the following sentences:

**The statistical comparison of the two instruments shows a positive bias of the Aeolus Rayleigh winds with respect to the A2D Rayleigh winds of 2.6 m·s⁻¹ and a standard deviation of 3.6 m·s⁻¹. Considering the accuracy and precision of the A2D wind data which was determined from the comparison with a highly-accurate coherent wind lidar as well as with ECMWF model winds, the systematic and random error of the Aeolus Rayleigh winds is determined to be 1.7 m·s⁻¹ and 2.5 m·s⁻¹, respectively.**

In the conclusions section, the following sentences were added:

**The statistical comparison revealed biases of -0.9 m·s⁻¹ and +1.6 m·s⁻¹ for the A2D and Aeolus LOS\* Rayleigh wind speeds with respect to the ECMWF model, respectively. Intercomparison of the two wind lidars showed a positive bias of the Aeolus Rayleigh winds with respect to the A2D of 2.6 m·s⁻¹, while the spreading between the two data sets of 3.6 m·s⁻¹ results from the respective random errors that add up quadratically. Considering the systematic and random error of the A2D, the accuracy of the Aeolus Rayleigh winds is determined to be +1.7 m·s⁻¹ and 2.5 m·s⁻¹ which is in line with the results from other validation studies performed for the commissioning phase of the Aeolus mission (Khaykin et al., 2019; Witschas et al., 2020).**

Comment #2.5:

*The discussion on the representability of the Aeolus cal/val results could be better developed in the context of the preliminary nature of L2B wind product.*

Response to Comment #2.5:

The following paragraph was added to the discussion of the lidar-lidar-comparison:

**In conclusion, due to the preliminary nature of the Aeolus L2B wind product, the Rayleigh winds exhibit relatively large systematic and random errors which are higher than the mission requirements (ESA, 2016). However, it should also be stated, that the representativity of the statistical results shown here is limited by the relatively small data set obtained from the WindVal III validation campaign. A strategy for increasing the number of compared winds, and hence the representativity of the Aeolus Cal/Val results in forthcoming campaigns is described in section 4.6.**